# An equivariant pretrained transformer for unified 3D molecular representation learning

Rui Jiao [1,2,6], Xiangzhe Kong [1,2,6], Li Zhang[1,2,6], Ziyang Yu[1,2], Fangyuan Ren [1,2,3], Wenjuan Tan[1,2], Wenbing Huang [4,5] ✉ & Yang Liu[1,2] ✉

Pretraining on a large number of unlabeled 3D molecules has showcased superiority in various scientific applications. However, prior efforts typically focus on pretraining models in a specific domain, missing the opportunity to leverage cross-domain knowledge. To mitigate this gap, we introduce Equivariant Pretrained Transformer, an all-atom foundation model that can be pretrained from multiple domain 3D molecules. Built upon an E(3)-equivariant transformer, the model learns both atom-level interactions and graph-level structural features (*e.g.* residuals in proteins), allowing it to generalize across diverse tasks. The model achieves strong gains in ligand binding affinity prediction, while also performing competitively in predicting properties of proteins and small molecules. We further show that the model can help identify potential antiviral compounds against the main protease of the COVID-19 virus, and validate promising candidates through computational and experimental studies.

Understanding and accurately representing the 3D geometric structure of molecular systems is critically important in numerous scientific domains, such as life sciences[1], drug discovery[2], and material design[3]. This significance arises because 3D structures largely determine molecular properties in downstream tasks, which cannot be effectively captured by simpler 1D representations like SMILES for chemical molecules or amino acid sequences for proteins. Recent advances in geometric Graph Neural Networks (GNNs)[4–6] have enabled the accurate modeling of physical interactions between atoms while respecting the E(3) symmetry inherent in molecular systems. These models incorporate 3D coordinates for each node, ensuring that scalar attributes and dynamic processes remain invariant or equivariant under E(3) transformations. Strategies such as irreducible representations[7], frame averaging[8], and scalarization mechanisms[4,5] have been employed to preserve the E(3) symmetry. Geometric GNN models have demonstrated remarkable success in various applications, including molecular property prediction[9], protein generation[10], and antibody design[11–13].

Despite remarkable successes, a central challenge remains: the scarcity of labeled data, which significantly limits the applicability of

deep learning methods in scientific research. To address this challenge, researchers have drawn inspiration from self-supervised pretraining techniques in natural language processing (NLP), as exemplified by BERT[14] and GPT[15]. Pretraining establishes a general-purpose 3D representation that captures atom-level interactions under E(3) symmetry. Downstream tasks such as property prediction or binding affinity estimation rely on accurate modeling of such interactions, which are implicitly learned during pretraining. Experimental results confirm that models pretrained on large-scale, unlabeled 3D molecular datasets exhibit substantial performance gains on downstream tasks after finetuning with limited labeled data. Specifically, for small molecules, approaches such as GraphMVP[16] and 3D Infomax[17] apply contrastive learning from 2D to 3D pairs, while MoleBlend[18] aligns 2D and 3D features through a multimodal pretraining framework. For proteins, GearNet[19] uses contrastive learning on sequential and structural representations. Other works, such as Uni-Mol[9], have explored cross-domain interactions, by pretraining separate models for small molecules and protein pockets before finetuning on binding datasets. In addition, inspired by score-based generative models,

[1]Department of Computer Science and Technology, Tsinghua University, Beijing, China. [2]Institute for AI Industry Research, Tsinghua University, Beijing, China. [3]Department of Pharmacy, University of Pisa, Pisa, Italy. [4]Gaoling School of Artificial Intelligence, Renmin University of China, Beijing, China. [5]Beijing Key Laboratory of Research on Large Models and Intelligent Governance, Beijing, China. [6]These authors contributed equally: Rui Jiao, Xiangzhe Kong, Li Zhang. ✉e-mail: hwenbing@ruc.edu.cn; liuyang2011@tsinghua.edu.cn

denoising tasks have emerged as powerful pretraining methods for learning force fields[20–23], with approaches like NERE[24] focusing on translation and rotation denoising at the ligand level.

Existing pretraining methods predominantly focus on domain-specific models, such as those tailored to small molecules[22,25], proteins[19], or dual-tower frameworks that separately handle small molecules and proteins[9,26]. However, these approaches are limited in their ability to generalize across multiple domains. We believe that developing a foundation model for diverse atomic systems is indispensable for advancing various scientific applications. On the one hand, recent breakthroughs in AI, such as the GPT series[15,27,28] and Gato[29], highlight the advantages of foundation models that unify learning across tasks and domains. These benefits include reduced need for handcrafted inductive biases, increased data diversity, and scalability with larger datasets and models. Extending this paradigm to molecular sciences could yield transformative results. On the other hand, from a physical perspective, all atomic systems, regardless of scale, are governed by the same fundamental principles, such as the Schrödinger Equation. Developing a single neural network capable of learning these universal principles represents a highly compelling pursuit.

Undoubtedly, building such a foundation model involves significant challenges that require addressing a range of technical and conceptual complexities. First, cross-domain data formats vary substantially. For small molecules, representations are typically at the atomic level, with each atom in a molecule corresponding to a specific set of features, such as atomic number, charge, and bond type. On the other hand, proteins are much more complex, requiring hierarchical representations where residues are composed of multiple atoms, and the structural context, including secondary structures, plays a critical role in determining function. This disparity in representation demands the development of a unified method capable of seamlessly integrating these different data formats and scales. Second, to capture the complex interactions between atoms and molecules, models must respect the fundamental symmetries that govern atomic systems. In particular, molecular interactions must be modeled while preserving E(3) symmetry, which accounts for the invariance of the system under rotations and translations in three-dimensional space. Finally, current self-supervised objectives used for pretraining are often task- or domain-specific. For example, objectives that work well for small molecules may not be directly applicable to proteins or other biological systems due to differences in their structural and functional properties.

In this paper, we propose an equivariant pretrained transformer (EPT), an all-atom foundation model that is pretrained from multiple-domain 3D molecules, including small molecules, proteins, and complexes. To our knowledge, we are one of the earliest attempts to develop a unified, all-atom representation model for handling diverse atomic systems across small molecules, proteins, and complexes in a self-supervised manner. Recent work, such as AlphaFold3[30], has also explored unified modeling of multiple molecular types, with a focus on accurate structure prediction across proteins, RNAs, and complexes. Firstly, by leveraging a unified molecular modeling component, EPT is able to not only process all-atom information but also incorporate block-level features that attend to a broader context of each atom, such as the atom-level surroundings for small molecules and the residue-level belongings for proteins. Secondly, EPT conforms to E(3) symmetry and is thoroughly designed upon a generic transformer. It derives the embedding layer with a one-layer equivariant GNN to reflect the graph geometry, and then updates the atom-level scalar and vector features via the equivariant self-attention and feed-forward mechanisms in each layer. Finally, the training objective requires the model to learn E(3)-equivariant denoising terms for reconstructing original conformations. This approach explicitly enables the model to not only understand the geometry of input systems, but also learn to stabilize noisy systems through rotation and translation. Notably, the

pretraining objectives are applied to entire blocks, rather than individual atoms, allowing the model to better capture hierarchical and contextual relationships.

To pretrain EPT, we construct a large-scale pretraining dataset of over 5.89M entries by integrating existing datasets from different sources. We then demonstrate the efficacy of EPT across diverse downstream tasks and datasets, achieving state-of-the-art results in several cases. Specifically, for ligand binding affinity (LBA) prediction, EPT outperforms existing methods on benchmarks with varying protein sequence similarities (30 and 60%). Furthermore, for the Mutation Stability Prediction (MSP) task, EPT showcases robust generalizability, significantly outperforming existing domain-specific models. For the molecular property prediction (MPP) task, EPT achieves competitive results compared to leading approaches, with significant gains when using block-level denoising strategies. These results validate EPT's ability to unify geometric learning across domains, providing a robust and generalizable framework for 3D molecular modeling. To further validate the utility of our EPT model, we apply EPT to rank FDA-approved drugs for their binding potential to the 3CL protease, a critical target in SARS-CoV-2 replication. The model successfully identifies known anti-COVID-19 drugs as top candidates, with further t-SNE analysis revealing that these drugs clustered closely with other high-affinity candidates predicted by EPT. From these clusters, we select 12 promising candidates for further validation through Molecular Dynamics (MD) simulations. Seven of these molecules exhibited higher binding affinities than the top-ranked known anti-COVID-19 drug. These results demonstrate the predictive power of EPT and its potential to accelerate drug discovery.

## Results
### Overview of EPT
EPT is an all-atom foundation model for multiple domains, including small molecules, proteins, and complexes (Fig. 1a). EPT creates a unified molecular representation across domains by defining "blocks" of atoms. For small molecules, each block is a collection of heavy atoms (non-hydrogen) and directly bonded hydrogens. For proteins, blocks are defined as amino acids. To enhance the model's ability to capture the block-level information, EPT utilizes a block-level pretraining strategy based on denoising (Fig. 1b). Specifically, each block is perturbed by a random translation of its center-of-mass (CoM) and a random rotation around the CoM. The model is trained to predict both the translational forces and angular momentum needed to restore the original structure, strengthening its ability to understand hierarchical geometries and maintain the physical integrity of molecular structures.

To effectively capture the geometry of molecular structures, EPT employs an improved transformer architecture that integrates E(3) symmetry as its backbone model (Fig. 1c). The process begins with a graph embedding layer that encodes both atom- and block-level information while initializing vector features through a message-passing layer. Subsequently, EPT alternates between equivariant self-attention and feed-forward layers to model interatomic relationships, producing rich scalar and vector representations. In total, EPT contains parameters exceeding 30M. Finally, the pretrained model is finetuned for various downstream tasks at the molecular, protein, and complex levels. Remarkably, EPT showcases strong effectiveness and interpretability in screening potential anti-COVID-19 drugs (Fig. 1d).

### Multi-domain pretraining dataset
We construct a large-scale pretraining dataset of over 5.89M entries by integrating existing 3D molecules from GEOM[31], PCQM4Mv2[32], PDB[33], and PDBBind[34]. For small molecules, we filter conformations with the top-5 Boltzmann weights for each molecule in GEOM[31], yielding 1.89M conformations. These are combined with 3.38M structures from PCQM4Mv2[32] to form the pretraining dataset. For proteins, we source

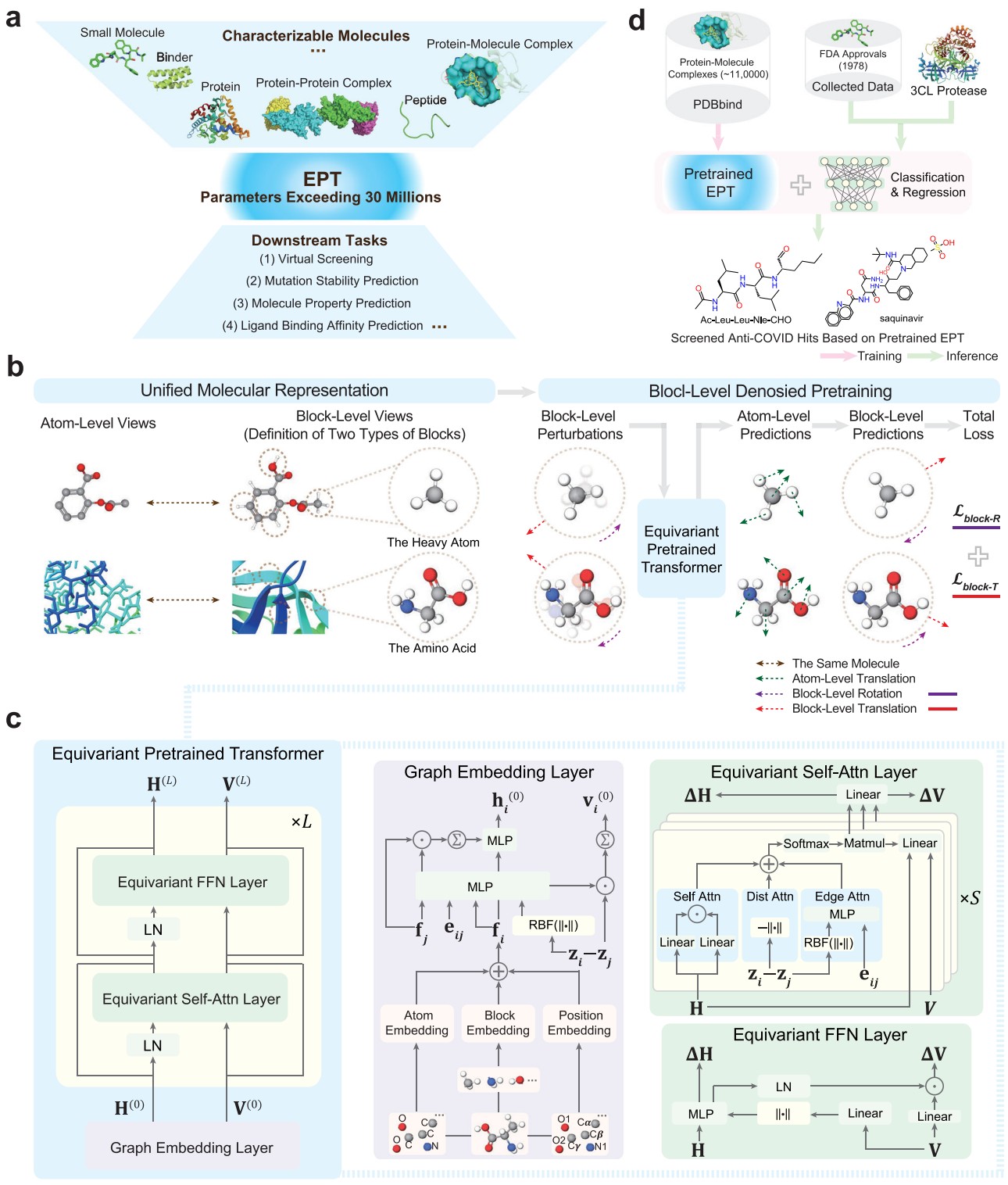

**Fig. 1 | Overview of EPT.** EPT is a foundation model for multi-domain 3D molecules. **a** EPT is capable of addressing diverse downstream tasks, after being pretrained on a large-scale hybrid dataset containing small molecules, proteins and complexes. **b** EPT integrates molecules from different domains by defining "blocks" as the fundamental units for each domain. For small molecules, blocks are defined as heavy atoms and their associated hydrogens, while for proteins, blocks correspond to amino acids. During pretraining, blocks are perturbed by random translations and rotations around the center-of-mass (CoM), and EPT is trained to recover the original structure. **c** In EPT, the atom representations including scalars **H** and vectors **V** are first initialized via a GNN-based embedding layer, and then updated by equivariant self-attention and feed-forward layers. **d** We demonstrate the efficacy of the pretrained EPT model in virtual screening for anti-COVID-19 drugs, outperforming computational and learning-based baselines.

experimental structures from the Protein Data Bank (PDB[33]), comprising 77,814 unique sequences along with 600k structural entries. Additionally, we incorporate 22,295 binding pockets from PDBBind[34] to enrich the dataset with information on protein–protein and protein–molecule interactions. During pretraining, we deliberately exclude all property labels to prevent data leakage and ensure unbiased performance evaluation in downstream tasks. The details of the pretraining dataset are illustrated in Fig. 5.

## Evaluation of model performance on downstream tasks

To evaluate the general performance of EPT, we select typical downstream tasks across two levels (Fig. 2a). At the binding level, we focus on ligand binding affinity prediction (LBA) and mutation stability prediction (MSP), while at the molecular level, we choose the molecule property prediction (MPP) task, with details as follows.

**Ligand binding affinity prediction (LBA).** We address the LBA task using the Atom3D dataset[35], which involves predicting the binding affinity between a protein pocket and its corresponding ligand. Following the setup in ref. 35, each sample in the dataset comprises a protein−molecule complex paired with its binding affinity value. The dataset offers two split configurations based on protein sequence similarity thresholds: ID30 (sequence identity capped at 30%) and ID60 (capped at 60%). These splits include 3507, 466, and 490 complexes in the training, validation, and test sets, respectively. For evaluation, we employ the root mean square error (RMSE), Pearson correlation coefficient, and Spearman correlation coefficient as metrics. To ensure the robustness of our results, we conduct experiments using three random seeds and report the mean and standard deviations for each metric. We compare our method against three categories of prior works: sequence-based methods, including DeepDTA[36], TAPE[37], and ProtTrans[38]; structure-based models, such as MaSIF[39], IEConv[40], Holoprot[41], ProtNet[42], and the three backbone models introduced by Atom3D[35]; and recent pretraining-based methods, including EGNN-PLM[43], Uni-Mol[9], and ProFSA[44]. To avoid implementation bias, we do not retrain these baselines, but directly report their performance from ProtNet and ProFSA, which follow the same dataset and split settings.

The results in Fig. 2b evaluate our EPT model under four different training conditions: trained from scratch, pretrained on the small-molecule subset, pretrained on the protein subset, and pretrained on the entire multi-domain dataset. We have the following observations: (1) Structure-based models generally surpass sequence-based counterparts, underscoring the significance of 3D geometry in capturing interactive information. (2) Pretraining on each individual subset is capable of enhancing performance. Remarkably, EPT-Molecule, which is pretrained without exposure to protein or complex structures, still outshines the scratch-trained model, suggesting the presence of cross-domain transferable knowledge. (3) Interestingly, even EPT-Scratch outperforms prior pretraining-based baselines such as Uni-Mol. This can be explained by architectural differences: Uni-Mol adopts a dual-tower structure that processes proteins and ligands separately, whereas EPT uses a unified model that jointly encodes the entire protein-ligand complex. These results actually align with our motivation of developing a unified model, which better captures the geometric nature of molecular complexes and enhances downstream performance. (4) EPT-MultiDomain, benefited from the entire dataset containing diverse domains, outperforms previous methods and achieves state-of-the-art performance on both of the splits. This implies that the breadth of pretraining data correlates positively with the model's performance, and enables a more generalizable understanding of biological interactions.

**Mutation stability prediction (MSP).** Additionally, we evaluate the model on the MSP task to assess its generalization capabilities. This task involves predicting whether a point mutation at the interface of protein complexes improves binding affinity, framed as a binary classification problem. The MSP dataset is derived from the SKEMPI 2.0 database[45], containing experimentally measured binding affinities of protein−protein complexes before and after single-point mutations. Each mutation is labeled as 1 if it increases binding affinity (i.e., $K_d$ of mutant < wild-type), and 0 otherwise. The dataset includes 893 positive and 3255 negative samples. Following ref. 35, we report AUROC scores on the dataset split with 30% sequence identity. This split comprises 2864 samples for training, 937 for validation, and 347 for testing.

We compare the scratch-trained EPT with Atom3D-CNN, GNN, and ENN[35], as well as GVP[46] and GearNet-Edge[19], as shown in Fig. 2c. Additionally, we evaluate the multi-domain pretrained EPT against prior structure-based pretraining strategies using GearNet-Edge[19], including Multiview Contrast, DiffPreT, and SiamDiff, as illustrated in Fig. 2d. The results demonstrate that EPT outperforms all baselines, both when trained from scratch and when pretrained on external dataset, highlighting its superior foundational expressiveness and exceptional knowledge transferability in modeling macromolecular systems.

**Molecule property prediction (MPP).** For the MPP task, we select QM9[47] to evaluate the performance of EPT on small molecules. In detail, QM9 serves as a quantum chemistry benchmark that offers 12 chemical properties for each 3D molecule composed of C, H, O, N, and F elements. Following ref. 48, we randomly select 10,000 and 10,831 structures for validation and testing, and the remaining 110,000 structures are used to finetune the model.

To benchmark EPT, we compare it against a variety of 3D geometric models and pretraining approaches designed for small molecules. For geometric GNNs, the evaluation includes SchNet[4], E(n)-GNN[5], DimeNet++[49], PaiNN[50], and Transformer-based architectures such as TorchMD-Net[48] and Equiformer[6], which leverage vector or higher-order features. Pretraining comparisons are drawn from ref. 23, including GeoSSL[21] and 3D-EMGP[22], which apply denoising techniques to PaiNN and E(n)-GNN, as well as Transformer-M[25], DP-TorchMD-Net[20], and Frad[23], which utilize various denoising strategies with transformer-based models. Model performance is evaluated using the mean absolute error (MAE) for each property, and the average rank across 12 tasks is computed to provide a concise summary of the results.

In Table 1, our EPT outperforms or matches the performance of existing denoising-based methods, underscoring the effectiveness of multi-domain block-level pretraining. Additionally, we evaluate an augmented model, **EPT-10**, which incorporates ten layers as opposed to the original six-layer configuration. The improved results, presented in the last row of Table 1, demonstrate that performance improves as the model complexity is scaled up.

**Analyses of the core components in EPT.** We provide a series of ablation studies on the MPP task (QM9 dataset) to elucidate the contribution of each component to the performance of our backbone model, as detailed in Fig. 3a. Specifically, we explore the following aspects. **1.** We first substitute the input embedding delineated in Eq. (5) with a straightforward atom-level embedding, denoted as $f_i = f_a(a_i)$. The findings suggest that enriching atom features with block-level information slightly improves model performance. **2.** In Eq. (10), we integrate the distance matrix **D** and edge features **R** into the attention mechanism. Eliminating either or both of these elements leads to a decline in performance, thereby underscoring their collective significance in effectively capturing the varying interatomic relations. **3.** In Eq. ((13–15)), we employ the FFN layers to amalgamate scalar and vector features. The resultant sharp drop after removing the FFN layers underscores the critical role of feature fusion in our model.

In addition, we evaluate the influence of pretraining datasets and denoising strategies on the LBA and MPP tasks in Fig. 3b. As an extension of Fig. 2b, we observe a consistent trend where pretraining on one domain confers benefits to downstream tasks in another domain. Specifically, the model pretrained on small molecules demonstrates enhanced performance on the LBA task, while the model pretrained on proteins exhibits improved results on the QM9 benchmark. In general, the model pretrained on the multi-domain dataset shows superior performance across all evaluated downstream tasks. We further compare the three kinds of denoising strategies introduced in SS3. The $\mathcal{L}_{atom}$ strategy focuses on atom-level denoising, showing superior results for small molecular structures. However, its benefits

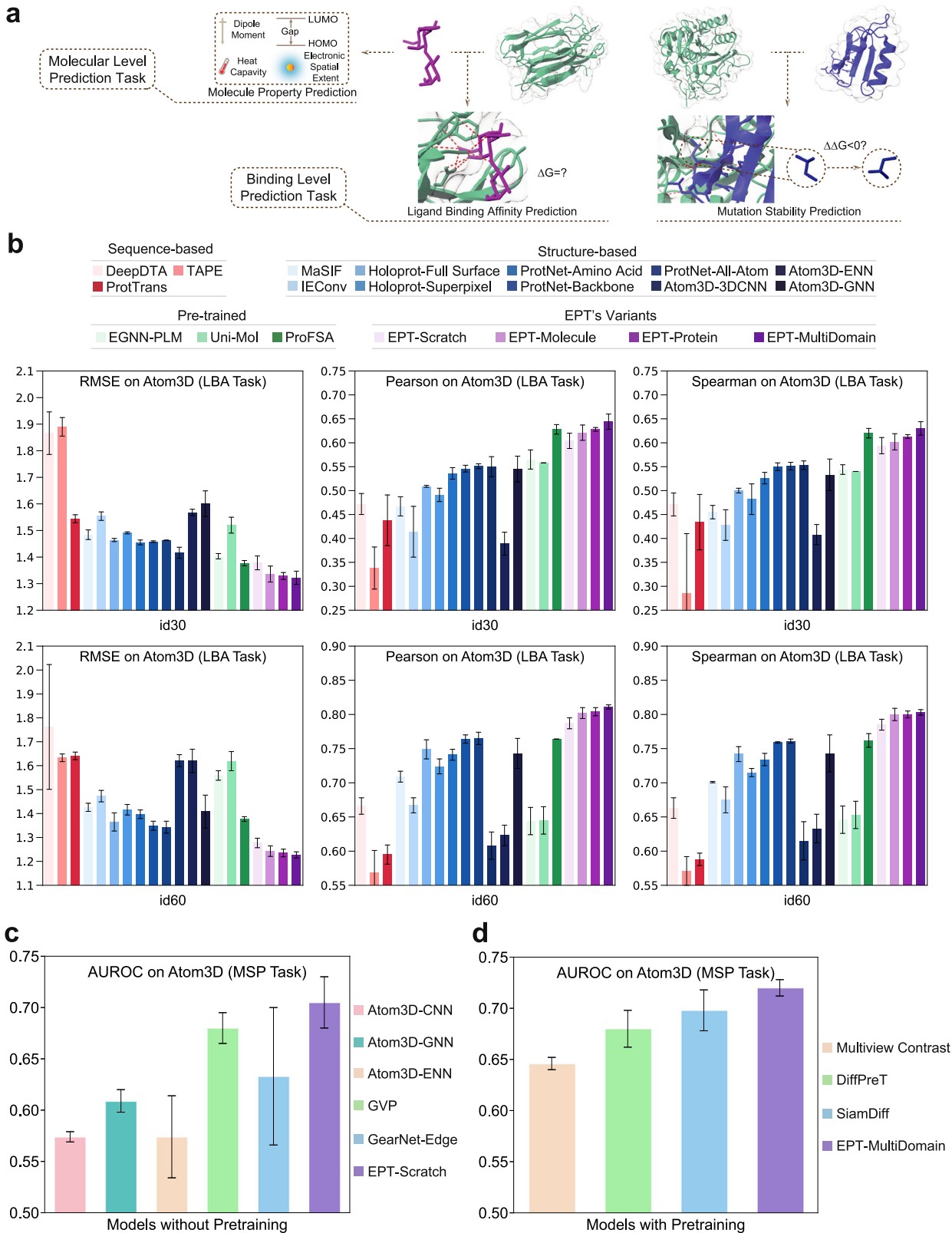

**Fig. 2 | The performance of EPT on downstream tasks. a** EPT is evaluated on three downstream tasks: ligand binding affinity prediction (LBA), mutation stability prediction (MSP) and molecule property prediction (MPP). **b** The root mean square error (RMSE), Pearson and Spearman correlation coefficients on both the id30 and id60 splits for the LBA task. **c** The AUROC on Atom3D for the MSP task by the models without pretraining. **d** The AUROC on Atom3D for the MSP task by the models with pretraining. Error bars are defined as the standard deviation of three runs.

**Table 1 | MAEs for the MPP task on the QM9 dataset**

| Model | $\mu \downarrow$ (D) | $\alpha \downarrow$ ($a_0^3$) | $\epsilon_{HOMO} \downarrow$ (meV) | $\epsilon_{LUMO} \downarrow$ (meV) | $\Delta \epsilon \downarrow$ (meV) | $\langle R^2 \rangle \downarrow$ ($a_0^2$) | ZPVE $\downarrow$ (meV) | $U_0 \downarrow$ (meV) | $U \downarrow$ (meV) | $H \downarrow$ (meV) | $G \downarrow$ (meV) | $C_v \downarrow$ ($\frac{cal}{mol K}$) | Avg. $\downarrow$ Rank |
|---|---|---|---|---|---|---|---|---|---|---|---|---|---|
| SchNet | 0.033 | 0.235 | 41.0 | 34.0 | 63.0 | 0.070 | 1.70 | 14.00 | 19.00 | 14.00 | 14.00 | 0.033 | 11.83 |
| E(n)-GNN | 0.029 | 0.071 | 29.0 | 25.0 | 48.0 | 0.110 | 1.55 | 11.00 | 12.00 | 12.00 | 12.00 | 0.031 | 11.17 |
| DimeNet++ | 0.030 | 0.043 | 24.6 | 19.5 | 32.6 | 0.330 | 1.21 | 6.32 | 6.28 | 6.53 | 7.56 | 0.023 | 7.17 |
| PaiNN | 0.012 | 0.045 | 27.6 | 20.4 | 45.7 | 0.070 | 1.28 | 5.85 | 5.83 | 5.98 | 7.35 | 0.024 | 6.33 |
| TorchMD-Net | 0.011 | 0.059 | 20.3 | 18.6 | 36.1 | 0.033 | 1.84 | 6.15 | 6.38 | 6.16 | 7.62 | 0.026 | 7.08 |
| Equiformer | 0.011 | 0.046 | 15.0 | 14.0 | 30.0 | 0.251 | 1.26 | 6.59 | 6.74 | 6.63 | 7.63 | 0.023 | 6.00 |
| Transformer-M | 0.037 | 0.041 | 17.5 | 16.2 | 27.4 | 0.075 | 1.18 | 9.37 | 9.41 | 9.39 | 9.63 | 0.022 | 6.92 |
| GeoSSL | 0.015 | 0.046 | 23.5 | 19.5 | 40.2 | 0.122 | 1.31 | 6.92 | 6.99 | 7.09 | 7.65 | 0.024 | 8.42 |
| 3D-EMGP | 0.020 | 0.057 | 21.3 | 18.2 | 37.1 | 0.092 | 1.38 | 8.60 | 8.60 | 8.70 | 9.30 | 0.026 | 8.83 |
| DP-TorchMD-Net | 0.012 | 0.052 | 17.7 | 14.3 | 31.8 | 0.450 | 1.71 | 6.57 | 6.11 | 6.45 | 6.91 | 0.026 | 6.67 |
| Frad | **0.010** | **0.037** | 15.3 | 13.7 | 27.8 | 0.342 | 1.42 | **5.33** | 5.62 | 5.55 | **6.19** | **0.020** | 3.17 |
| EPT | 0.011 | 0.045 | 16.2 | 14.1 | 29.6 | 0.122 | 1.14 | 5.53 | 5.70 | 5.52 | 6.42 | **0.020** | 3.33 |
| EPT-10 | **0.010** | 0.045 | 15.2 | **13.6** | 29.0 | 0.152 | **1.11** | **5.44** | **5.54** | **5.42** | 6.37 | **0.020** | **2.33** |

The best results are in **bold** and the second best are underlined. The right-most column provides the averaged rank of each method across 12 tasks. EPT-10 denotes a ten-layer variant of EPT to evaluate model scalability. We also include the performance for both Frad and EPT when calling the networks from 6 to 12 layers in the Supplementary Fig. 5. Frad does not benefit consistently from a larger network size, while EPT continues to improve up to 10-12 layers, indicating better scalability and stability.

are less pronounced when applied to large, complex systems such as the protein–molecule complexes in LBA. On the contrary, $\mathcal{L}_{block-T}$ adopts a more macroscopic approach by only considering the translations of each block's center-of-mass. This coarse-grained strategy improves performance for larger systems but tends to struggle in smaller molecules. Finally, our model, which accounts for both block-level rotations and translations, provides comprehensive supervision for predicted forces, resulting in the optimal performance across all strategies.

**EPT-accelerated discovery of anti-COVID-19 candidates**

We conduct virtual screening experiments to find potential anti-COVID-19 small-molecule drugs. To this end, we design benchmark experiments using the PDBBind dataset, which is partitioned into three subsets based on different strategies: id30, id60, and scaffold splits. For each partition, we define two evaluation tasks, namely the affinity prediction task and the ranking task. The affinity prediction task involves predicting the binding affinity of all ligand-pocket complexes and measuring the Spearman and Pearson correlations between the model's predictions and the ground-truth affinity labels. The ranking task aims to determine whether a given ligand candidate can bind to the target pocket. To achieve this, we randomly sample 10 additional ligands as negative examples for each pocket and train a classification head to distinguish between positive and negative ligands. We use the Glide program for docking negative samples. The classifier outputs are then used to rank the 11 samples (1 positive and 10 negatives). The model's performance is evaluated using two metrics: the average rank of the positive sample (AvgRank) and the proportion of cases where the positive sample is ranked as the top candidate (Top-1 Accuracy).

We adopt the pretrained EPT and finetune it on the training complexes. To benchmark EPT, we compare its performance against two typical baselines: the classical method, Glide[51] based on energy computation, and the dual-tower model ESM+Uni-Mol, which applies the pretrained protein model ESM[52] for pocket modeling and Uni-Mol[9] for representing small molecules. As shown in Fig. 4a, our model achieves significantly better performance than both Glide and the dual-tower models in the affinity prediction tasks. For ranking tasks, EPT outperforms the dual-tower models and performs on par with Glide. These results underscore the effectiveness of our unified representation model in capturing the complex interactions between ligands and their target protein pockets. Note that we limited the negative ligands to 10 per target across all 500 targets. While we recognize that virtual screening typically involves ranking binders from much larger compound libraries, scaling to millions of molecules across 500 targets would be computationally prohibitive. Crucially, our further analysis here shows that the evaluation metric remains stable whether using 10 or more decoys per target (Supplementary Fig. 7), demonstrating that this reduced setting still enables robust performance comparisons across targets.

Based on these findings, we apply our finetuned model to rank 1978 FDA-approved drugs (including 26 antiviral drugs and eight marketed anti-COVID-19 drugs) in order to evaluate their binding potential to 3CL protease, a key target in SARS-CoV-2 replication. The rankings of 8 anti-COVID-19 drugs are shown in Fig. 4b, where we observe that nearly all drugs are ranked within the top 200. We visualize the representations of all ligand-pocket complexes using t-SNE in Fig. 4b. The results reveal that, with the exception of the structurally distinct outlier Xocova, which features a non-peptidic triazine scaffold (see Supplementary Fig. 6), anti-COVID-19 drugs generally form a concentrated cluster surrounded by other drugs with high EPT ranking scores. Additionally, we select the top six antiviral drugs based on their rankings and further identify the top six non-antiviral, non-anti-COVID-19 drugs as extra potential anti-COVID-19 candidates. These 12 top-ranking candidates are located close to seven marketed anti-COVID-19 drugs in the t-SNE embeddings shown in Fig.

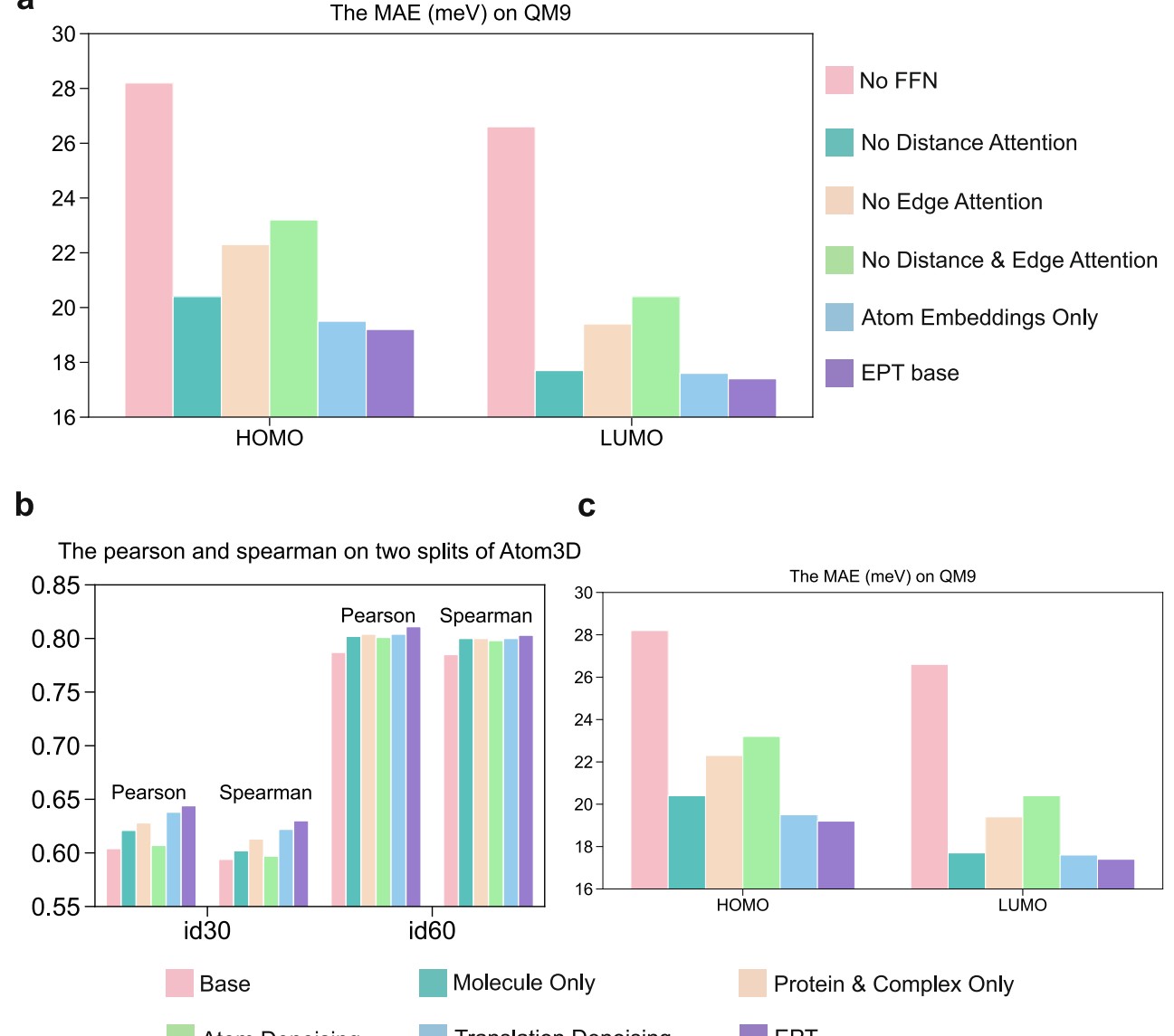

**Fig. 3 | The validation for EPT's core modules and pretraining strategy. a** Mean absolute errors (MAE) of the HOMO and LUMO predictions on QM9, under different EPT variants. **b** Pearson and Spearman correlation coefficients on id30 and id60 splits for the LBA task, under different pretraining strategies. "Base" refers to the EPT model without pretraining. **c** MAE of the HOMO and LUMO predictions on QM9, under different pretraining strategies.

4b, indicating their potential high affinity for the 3CL protease. To quantitatively assess the binding potential of the 12 selected candidates, we perform molecular docking and Molecular Dynamics (MD) simulations, identifying two hits, Ac-Leu-Leu-Nle-CHO and Saquinavir, with EPT-predicted dissociation constants ($K_d$) of 87.4 and 24.5 nM, respectively. Detailed protocols for molecular docking, MD simulations, and $\Delta G$ calculation are provided in the Supplementary Notes 4.2 and 4.3. The SARS-CoV-2 3CL pro-ligand complexes were simulated using GROMACS 2024.2[53]. Energy minimization was performed using the steepest descent algorithm, followed by 100 ps of restrained equilibration under the constant number of particles, volume, and temperature (NVT) ensemble and 100 ps under the constant number of particles, pressure, and temperature (NPT) ensemble. Subsequently, 100 ns of unrestrained production MD simulations were carried out at 310 K and 1 bar. As exhibited in Fig. 4c, d, on one hand, the docking results indicate that these two hits indeed have some interactions, such as hydrogen bonds and hydrophobic interactions with 3CL protease that enhance binding affinities. On the other hand, compared to the positive reference Leritrelvir with $\Delta G$ of − 6.16 kcal/mol, $\Delta G$ of

these two hits are both ∼−30 kcal/mol, further suggesting their potential as clinical drugs. Additional examples can be found in Supplementary Fig. 2. Of the 12 selected candidates, 9 demonstrate binding potential to the target with $\Delta G < 0$, and 7 deliver lower $\Delta G$ than Leritrelvir, the top-ranking anti-COVID-19 drug in Fig. 4b. These results not only emphasize the predictive power of EPT but also highlight its potential to accelerate the identification of promising drug candidates.

Among the two strong hits, Saquinavir (Fig. 4d) is reported by the literature to inhibit 3CL protease with an IC50 of 9.92 μM[54]. We further tested Ac-Leu-Leu-Nle-CHO (Fig. 4c), which showed an $IC_{50}$ of 5.47 μM (see Supplementary Fig. 4). These results confirm their inhibitory potency. Full details are provided in the Supplementary Note 5.

## Discussion
In this work, we propose EPT, an equivariant transformer-based model pretrained on multi-domain 3D molecular structures. We unify the representation of molecules from different domains by introducing the concept of blocks, enabling the capture of hierarchical information from atomic-level details in small molecules to residue-level features in

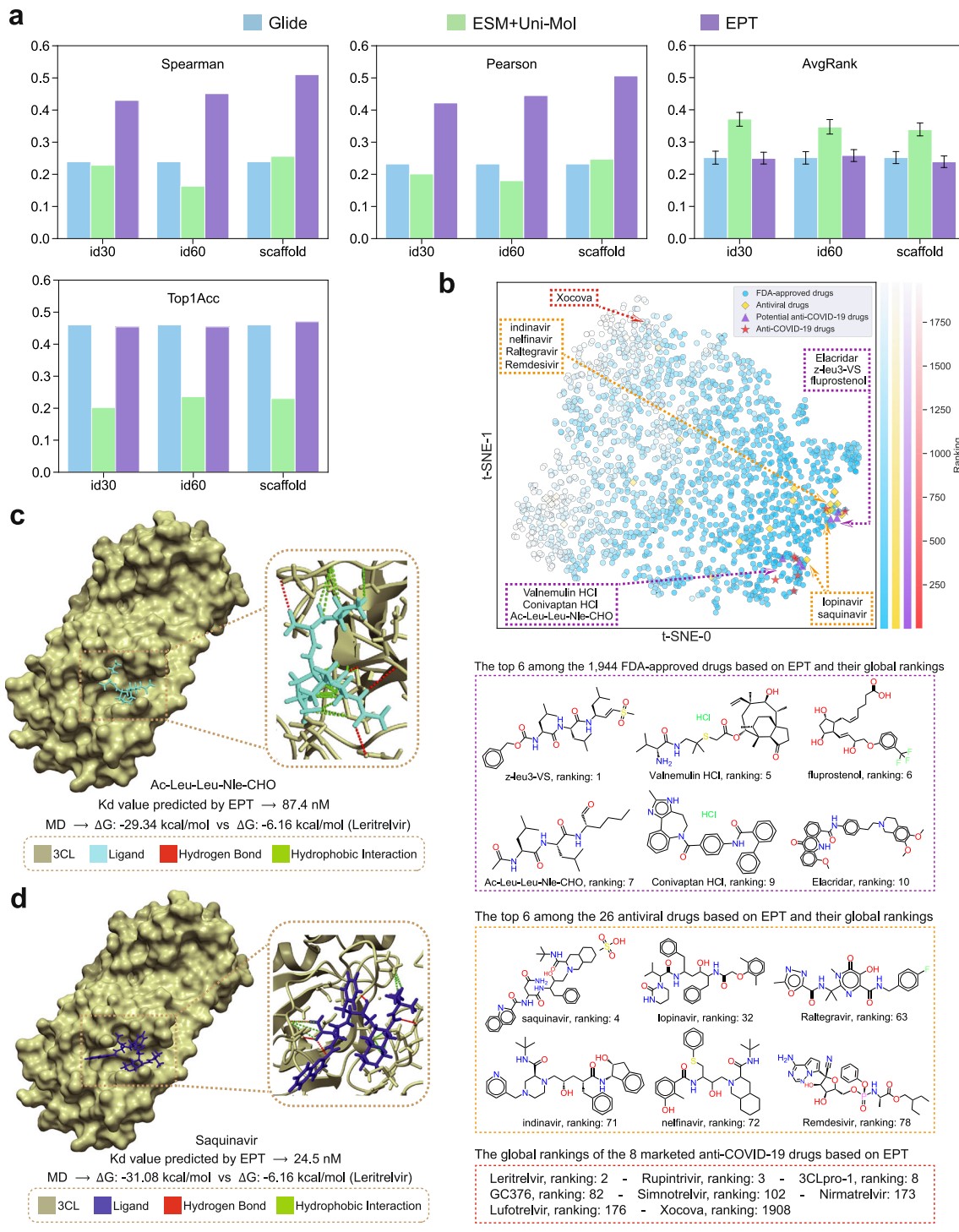

**Fig. 4 | The application of EPT in screening anti-COVID-19 drugs. a** Benchmark experiments on the PDBBind dataset: predicting the binding affinity (evaluated by the Pearson and Spearman correlation coefficients), and identifying the positive candidate bound to the target pocket (evaluated by the ranking metrics AvgRank and Top1Acc). The error bar of AvgRank is calculated as the standard deviation of all pockets. **b** Visualization of EPT embeddings, rankings of 1978 FDA-approved drugs based on EPT, and the presentation of focus molecules. **c, d** A comprehensive analysis of the two hits screened based on EPT is conducted through molecular docking and MD simulations. Leritrelvir is a positive reference.

proteins. Leveraging this, we design a block-level denoising pretraining strategy, where the model learns to recover from geometric perturbations applied to these blocks. To effectively handle this task, we utilize an equivariant Transformer architecture that preserves the intrinsic symmetries of physical laws, ensuring that the model captures the complex geometric relationships while being scalable for large, diverse datasets.

The effectiveness of EPT is demonstrated through its superior performance on a variety of benchmark tasks, including ligand binding affinity (LBA) prediction, mutation stability prediction (MSP), and molecular property prediction (MPP). EPT outperforms existing domain-specific methods in these tasks, highlighting its ability to generalize across different molecular domains. Furthermore, when applied to predict the binding potential of FDA-approved drugs to the

3CL protease, a critical target in the fight against SARS-CoV-2, EPT identified known anti-COVID-19 drugs as top candidates and facilitated the identification of promising drug candidates, showcasing its real-world applicability.

The development of EPT opens exciting possibilities for future research and applications in molecular science. From the model design perspective, recent studies on unified modeling for atomic systems spans several directions, including structure aware generalist encoders learning transferable 3D representations across molecules[9,55], unified 3D generative models across different domains[30,56,57], and sequence level foundation models from language-model-based paradigms[58-60]. We provide more discussions in Supplementary Note 11. EPT currently focuses on representation learning and predictive understanding, and aligns most closely with structure-aware encoders. Nevertheless, its block-aware, E(3)-equivariance nature also makes it well-suited as a backbone for generative models[61] and as a geometric encoder for multimodal sequence-based models[62], which could be promising future directions. Moreover, future work could also involve extending the model to other molecular systems like crystals, catalysts, or nucleic acids, and further enhancing its denoising techniques to handle even more complex molecular interactions. We generalize EPT on RNA-ligand affinity prediction in Supplementary Note 6 as an initial exploration. However, it currently depends on vocabulary extensions to decompose unseen molecular systems into blocks. A potential avenue for improvement is to alleviate this reliance by leveraging data-driven block discovery[63]. The success of EPT also paves the way for future foundation models in molecular science, where unified, scalable models can drive breakthroughs across a broad range of scientific domains, from drug design to protein engineering.

## Methods
### Datasets
We curate a comprehensive structural dataset comprising 5.89 million samples and 261 million blocks, spanning small molecules, proteins, and complexes (Fig. 5a). Among these, small-molecule datasets, including GEOM[31] and PCQM4Mv2[32], contribute 35.8% of the total blocks. For protein structures, experimental data are collected from the Protein Data Bank (PDB[33]), which encompasses 600,000 structural entries and accounts for 59.4% of the blocks. The remaining 4.8% of blocks are derived from PDBBind[34], including protein−protein complexes (PP), protein−molecule complexes from the refined set, and other protein−molecule complexes collected before version 2020, denoted as v2020-other-PL. Each sample is stored as a sequence of blocks, with each block defined by four attributes: the types of atoms within the block, the type of the block itself, the position index of each atom, and the 3D coordinate matrix of the block.

We employ virtual screening as a typical application of the pre-trained EPT model (Fig. 5b). To finetune EPT, we first prepare the Docked-PDBBind dataset. Specifically, we obtain 10,000 protein-small-molecule complexes from PDBBind and re-dock them using the Glide software[51] to generate 10,000 positive samples. Additionally, for each complex, we randomly selected ten non-binding small molecules and docked them with the protein using Glide, yielding 100,000 negative samples. The re-docking process ensures consistency with the inference phase, where only individual protein and small-molecule structures are available, rather than pre-formed complexes. This alignment is crucial to maintain distributional similarity between the training data and the docking results used during inference. For inference, we dock the target 3CL protease with 1978 FDA-approved drugs and rank them based on the predicted binding probabilities from the finetuned model.

### Model architecture
We represent a molecule with $N$ atoms as a fully connected graph $\mathscr{G}$, where atoms are depicted as nodes, and their interactions are depicted as edges $\mathscr{E}$. To capture the high-level structure within molecules,

atoms are grouped into $M$ predefined blocks to enrich the node features[64]. Let $m_i$ denote the index of the block containing atom $i$, the feature set for an atom is extended to $(a_i, b_{m_i}, p_i, \mathbf{z}_i)$, where $a_i \in \mathscr{A}$ specifying the atom type, $b_{m_i}$ indicates the block type of $m_i$, $p_i$ denotes the atom's positional index within its block, and $\mathbf{z}_i \in \mathbb{R}^3$ represents the atom's 3D coordinate. The positional index within an amino acid is determined using Greek letters, ordered based on their distance from the $C_\alpha$ atom. For small molecules, the positional index is represented by a designated token, [Mol]. While $\mathscr{G}$ is fully connected in the sense that all atom pairs are processed by the self-attention mechanism, we assign discrete edge types based on geometric distance and topology to enhance local interaction modeling. Detailedly, local interactions between atoms are categorized into three distinct edge types to reflect both intra-block and inter-block relationships. Mathematically,

$$e_{ij} = \begin{cases} 0, & m_i = m_j, \\ 1, & m_i \neq m_j, d(m_i, m_j) \leq \delta_{\text{topo}}, \\ 2, & m_i \neq m_j, \delta_{\text{topo}} < d(m_i, m_j) \leq \delta_{\max}, \end{cases} \quad (1)$$

where $\delta_{\text{topo}}$ and $\delta_{\max}$ are predefined thresholds that represent topological and maximum allowable distances, respectively. The function $d(m_i, m_j) = \min_{m_p = m_i, m_q = m_j} \| z_p - z_q \|_2$ calculates the minimum Euclidean distance between any two atoms belonging to blocks $m_i$ and $m_j$.

In the pursuit of efficiently capturing the nuanced interactions of atoms within molecules, we present the equivariant full-atom transformer. It utilizes the transformer-based backbone[65] to model the complex interactions among atoms, while updating the scalar and vector features to capture the rich geometric information inherent in molecular structures.

Our model first acquires the input features from the Graph Embedding layer, and iteratively updates the features at each layer $l$. Let $\mathbf{H}^{(l)} = [\mathbf{h}_1^{(l)}, \mathbf{h}_2^{(l)}, \cdots, \mathbf{h}_N^{(l)}] \in \mathbb{R}^{h \times N}$ denote the scalar, and $\mathbf{V}^{(l)} = [\mathbf{v}_1^{(l)}, \mathbf{v}_2^{(l)}, \cdots, \mathbf{v}_N^{(l)}] \in \mathbb{R}^{3 \times h \times N}$ denote the vector. The model is constructed in this way:

$$[\mathbf{H}^{(0)}, \mathbf{V}^{(0)}] = \text{Embedding}(\mathbf{A}, \mathbf{B}, \mathbf{P}, \mathbf{Z}), \quad (2)$$

$$[\mathbf{H}^{(l-0.5)}, \mathbf{V}^{(l-0.5)}] = [\mathbf{H}^{(l-1)}, \mathbf{V}^{(l-1)}] + \text{Self} - \text{Attn}(\text{LN}(\mathbf{H}^{(l-1)}), \mathbf{V}^{(l-1)}), \quad (3)$$

$$[\mathbf{H}^{(l)}, \mathbf{V}^{(l)}] = [\mathbf{H}^{(l-0.5)}, \mathbf{V}^{(l-0.5)}] + \text{FFN}(\text{LN}(\mathbf{H}^{(l-0.5)}), \mathbf{V}^{(l-0.5)}). \quad (4)$$

After the embedding layer, self-attention (Self-Attn) and feed-forward networks (FFN) are applied alternately, with pre-layer normalization (LN) and residual connections preceding each operation. We modify the Self-Attn and FFN layers to be E(3)-equivariant, preserving the geometrical symmetry of molecular structures. These layers are detailed as follows. For conciseness, we omit the layer subscribe $l$ unless otherwise specified.

For the graph embedding layer, the input features are obtained as:

$$\mathbf{f}_i = f_b(b_{m_i}) + f_a(a_i) + f_p(p_i), \quad (5)$$

$$\mathbf{e}'_{ij} = [\mathbf{f}_i, \mathbf{f}_j, \mathbf{e}_{ij}, \text{RBF}(\| \mathbf{z}_i - \mathbf{z}_j \|_2)], \quad (6)$$

$$\mathbf{h}_i^{(0)} = \varphi_h(\mathbf{f}_i, \sum_{j \in \mathscr{N}(i)} \varphi_s(\mathbf{e}'_{ij}) \cdot \mathbf{f}_j), \quad (7)$$

$$\mathbf{v}_i^{(0)} = \sum_{j \in \mathscr{N}(i)} \varphi_v(\mathbf{e}'_{ij}) \cdot (\mathbf{z}_i - \mathbf{z}_j), \quad (8)$$

where $f_b, f_a, f_p$ separately embed the block types, atom types and atom orders, RBF($\cdot$) denote the radial basis functions, and $\varphi_h$, $\varphi_s$, $\varphi_v$ are

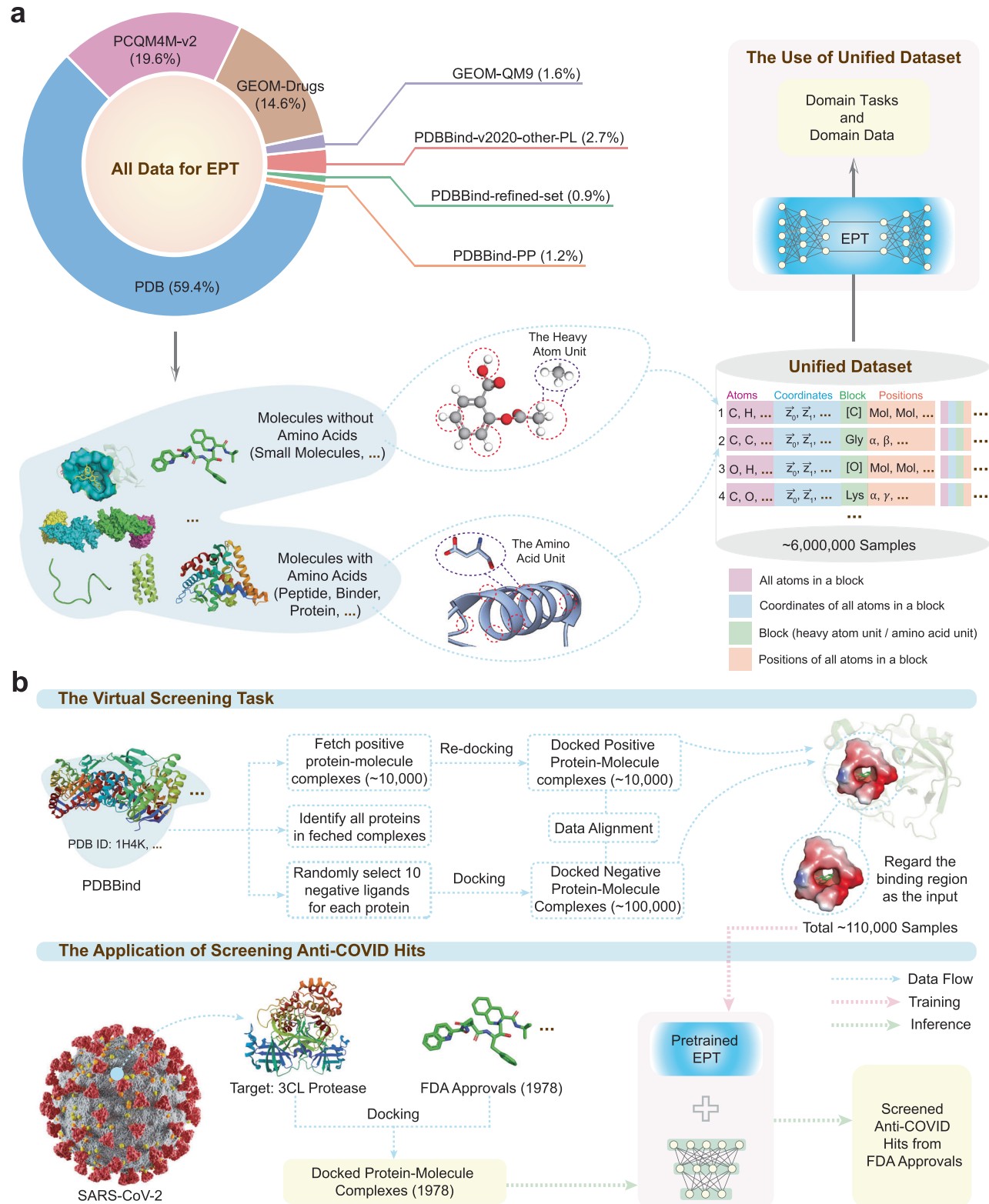

**Fig. 5 | Data description and preprocessing. a** The pretraining dataset combines small-molecule conformations, protein structures, as well as protein−protein and protein−molecule complexes. Each entry is represented as a list of blocks, with each block characterized by four features: the atom types, the block type, the ordered position indexes, and the atom coordinates. **b** We illustrate the overall pipeline for evaluating the virtual screening task. Fine-tuning of the pretrained EPT model is performed using a curated dataset of protein-small-molecule complexes collected from PDBBind and re-docked to generate positive and negative samples. The finetuned model is then applied to rank FDA-approved drugs based on their predicted binding probability from the docked complex with the 3CL protease.

MLPs to aggregate neighbor information to enrich the 0th layer features. It is necessary to initialize $\mathbf{V}^{(0)}$ with SE(3)-equivariant non-zero values via Eq. (8), otherwise the vector features will remain zeros in subsequent layers.

The self-attention layer plays a crucial role in modeling interatomic interactions. For each layer, query $\mathbf{Q}_s$, key $\mathbf{K}_s$, and value $\mathbf{V}_s$ matrices of the $s$-th head are computed as

$$\mathbf{Q}_s = \mathbf{H}\mathbf{W}_s^Q, \mathbf{K}_s = \mathbf{H}\mathbf{W}_s^K, \mathbf{V}_s = [\mathbf{H}\mathbf{W}_s^{Vh}, \mathbf{V}\mathbf{W}_s^{Vv}], \tag{9}$$

$\mathbb{R}^{h \times h_s}$ are trainable parameters that map the features to the appropriate query, key, and value spaces. And the attention mechanism is given by

$$\mathbf{H}_s, \mathbf{V}'_s = \mathrm{Softmax}\left(\frac{\mathbf{Q}_s^\top \mathbf{K}_s}{2\sqrt{h_s}} - \mathbf{D} + \mathbf{R}\right)\mathbf{V}_s, \tag{10}$$

where $\mathbf{D} = \{d_{ij}\}_{i,j=1}^N = \{\|\mathbf{z}_i - \mathbf{z}_j\|_2\}_{i,j=1}^N$ is the distance matrix, and $\mathbf{R} = \{r_{ij}\}_{i,j=1}^N$ encodes the edge interactions and geometric relations as

$$r_{ij} = \begin{cases} \varphi_r(\mathbf{e}_{ij}, \mathrm{RBF}(\|\mathbf{z}_i - \mathbf{z}_j\|_2)), & (i,j) \in \mathscr{E}, \\ 0, & (i,j) \notin \mathscr{E}. \end{cases} \tag{11}$$

Here $\varphi_r$ is an MLP. The outputs of the self-attention layer combines the contributions of all heads:

$$\Delta\mathbf{H} = \sum_s \mathbf{H}_s \mathbf{W}_s^{Oh}, \Delta\mathbf{V} = \sum_s \mathbf{V}'_s \mathbf{W}_s^{Ov}, \tag{12}$$

where $\mathbf{W}_s^{Oh}, \mathbf{W}_s^{Ov} \in \mathbb{R}^{h_s \times h}$ are head-specific trainable parameters.

Building upon the Geometric Vector Perceptron (GVP[46]) concept, the equivariant feed-forward layer is where the scalar and vector features are fused and updated simultaneously:

$$\mathbf{V}_1, \mathbf{V}_2 = \mathbf{V}\mathbf{W}_1, \mathbf{V}\mathbf{W}_2, \tag{13}$$

$$\Delta\mathbf{H}, \mathbf{U} = \varphi_{\mathrm{FFN}}(\mathbf{H}, \|\mathbf{V}_1\|_2), \tag{14}$$

$$\Delta\mathbf{V} = \mathrm{LN}(\mathbf{U}) \odot \mathbf{V}_2, \tag{15}$$

where $\mathbf{W}_1, \mathbf{W}_2 \in \mathbb{R}^{h \times h}$ are learnable linear projectors, $\varphi_{\mathrm{FFN}}$ is an MLP that integrates the scalar features with the magnitude of the vector features, and $\odot$ denotes element-wise multiplication. The intermediate matrix $\mathbf{U}$ is layer-normalized to conserve the scale of the updated vectors.

### Block-level denoised pretraining

Based on the backbone model above, an additional force prediction head is required for denoised pretraining. In practice, we apply an additional FFN-like layer over layer-L to fuse the output scalars and vectors as

$$\mathbf{H}_{\mathrm{out}}, \mathbf{V}_{\mathrm{out}} = \mathbf{H}^{(L)}, \mathbf{V}^{(L)} / \|\mathbf{V}^{(L)}\|_2, \tag{16}$$

$$\mathbf{F}' = \varphi_{\mathrm{out}}(\mathbf{H}_{\mathrm{out}}, \|\mathbf{V}_{\mathrm{out}}\mathbf{W}'_1\|_2) \odot \mathbf{V}_{\mathrm{out}}\mathbf{W}'_2. \tag{17}$$

To encode hierarchical molecular information, we introduce a block-level denoising pretraining strategy that integrates both translational and rotational perturbations applied to molecular blocks. This approach considers blocks as rigid bodies, where all atoms within a block are perturbed together, effectively capturing both intra- and inter-block geometric relationships.

Perturbations are applied by introducing translational and rotational noise to each block, modeled as:

$$\mathbf{Z}' = C\left(g_b(\mathbf{Z}_b + \sigma_t \boldsymbol{\epsilon}_{\mathbf{Z}_b}) + \mathbf{Q}_b\mathbf{Z}_r\right), \tag{18}$$

where $\boldsymbol{\epsilon}_{\mathbf{Z}_b} \sim \mathcal{N}(0, \mathbf{I}_{3M})$ represents translational noise applied at the block level, $g_b(\cdot)$ maps block coordinates to atoms, $\mathbf{Q}_b$ is the rotation matrix derived from rotational noise $\boldsymbol{\omega}_b$ sampled from the isotropic Gaussian distribution $\mathscr{IG}_{SO(3)}(\sigma_r)$[66], and $\mathbf{Z}_r = \mathbf{Z} - g_b(\mathbf{Z}_b)$ represents the relative atom positions to block centers. The centering operator $C(\cdot)$ ensures translational neutrality.

The pretraining objective incorporates two components. First, a translation objective minimizes the discrepancy between predicted resultant forces and translational noises averaged over blocks:

$$\mathscr{L}_{\mathrm{block-T}} = \mathbb{E}_{\boldsymbol{\epsilon}_{\mathbf{Z}_b} \sim \mathcal{N}(0, \mathbf{I}_{3M})}\left[\|\mu_b(\mathbf{F}') - \frac{\mu_b(\mathbf{Z}') - \mathbf{Z}_b}{\sigma_t^2}\|_2^2\right]. \tag{19}$$

where $\mu_b(\cdot)$ computes block-level averages. Second, a rotation objective aligns angular accelerations derived from predicted torques with gradients of the rotational noise distribution.

$$\mathscr{L}_{\mathrm{block-R}} = \mathbb{E}_{\boldsymbol{\omega} \sim \mathscr{IG}_{SO(3)}(\sigma_r)}\left[\|\boldsymbol{\alpha}_b - \nabla_{\boldsymbol{\omega}} p(\boldsymbol{\omega})\|_2^2\right]. \tag{20}$$

where $\boldsymbol{\alpha}_b = \mathbf{I}_b^{-1}\mathbf{M}'_b$ is the predicted angular acceleration, $\mathbf{M}'_b = \sum_j \left(\mathbf{z}_j - g_b(\mathbf{Z}_b)\right) \times \mathbf{f}'_j$ represents the torque, and $\mathbf{I}_b$ is the inertia matrix of each block. The combined block-level denoising loss integrates these two objectives:

$$\mathscr{L}_{\mathrm{block-C}} = \mathscr{L}_{\mathrm{block-T}} + \mathscr{L}_{\mathrm{block-R}}. \tag{21}$$

### Training regimen

We first pretrain the backbone model on the collected multi-domain dataset based on the objective $\mathscr{L}_{\mathrm{block-C}}$, leveraging 8 NVIDIA Tesla A800 GPUs. The pretraining process spans 50 epochs, incorporating translational and rotational noise scales of $\sigma_t = 0.04$ and $\sigma_r = 0.1$, respectively. A dynamic batching strategy is applied, capping each mini-batch at a maximum of 10,000 atoms per GPU. After pretraining, the force prediction head is omitted, and the backbone model is further finetuned with task-specific heads.

For MPP, we utilize the noisy node technique[67] by adding $\mathscr{L}_{\mathrm{block-C}}$ as an auxiliary training objective, and the entire loss for finetuning on QM9 can be formulated as $\mathscr{L} = \mathscr{L}_{\mathrm{MAE}} + \lambda\mathscr{L}_{\mathrm{block-C}}$, where $\lambda = 0.1$ balances the weight of each term.

For LBA, we apply a local mask where only attentions from neighbors within $\delta_{\max}$ of each node are calculated, and we consider three types of output heads based on an MLP $\varphi_E$ as follows:

$$\varphi_{\mathrm{atom}}(\mathbf{H}^{(l)}) = \sum_i \varphi_E(\mathbf{h}_i^{(l)}), \tag{22}$$

$$\varphi_{\mathrm{block}}(\mathbf{H}^{(l)}) = \sum_{m_i} \varphi_E\left(\sum_{m_j = m_i} \mathbf{h}_j^{(l)}\right), \tag{23}$$

$$\varphi_{\mathrm{graph}}(\mathbf{H}^{(l)}) = \varphi_E\left(\sum_i \mathbf{h}_i^{(l)}\right). \tag{24}$$

For MSP, we use the split by sequence identity over 30% provided by Atom3D[35], and extract all residues within 6 Å distance to the mutation point as the local view for input, where the distance between two residues is measured by the minimum distance between atom pairs.

For the virtual screening benchmark, we design two tasks. For the ranking task, we use a negative rate of 0.6 to adjust the fraction of negative data pairs during training. Additionally, for the prediction task, we use the same local mask as the LBA task. We list all hyperparameters in Supplementary Note 3 Implementation details.

## Evaluation metrics

Following Atom3D benchmark[35], we use mean absolute error (MAE) on the MPP task by measuring the average deviation between predicted properties and ground-truth labels. For the LBA task, we report RMSE, Pearson, and Spearman correlation coefficients to comprehensively evaluate model performance. RMSE (root mean square error) measures the absolute difference between predicted and ground-truth binding affinities, directly reflecting the accuracy of the predicted magnitudes. Pearson correlation coefficient (Pearson or PCC) assesses the linear relationship between predicted and actual values, focusing on the consistency of relative magnitudes, and thus is insensitive to systematic offsets or scale mismatches. Spearman correlation coefficient (Spearman or SPCC), on the other hand, captures the monotonic rank-order agreement between predictions and ground truth, evaluating how well the predicted affinities preserve the correct ordering. Together, these metrics offer complementary perspectives, respectively emphasizing numerical precision, linear alignment, and ranking accuracy. For the MSP task, we compute the area under the receiver operating characteristic curve (AUROC) to measure binary classification performance.

## Data availability

The raw datasets for pretraining and downstream tasks are available publicly at the following links: GEOM[31]: https://dataverse.harvard.edu/dataset.xhtml?persistentId=doi:10.7910/DVN/JNGTDF. PCQM4Mv2[32]: https://ogb.stanford.edu/docs/lsc/pcqm4mv2. PDB[33]: https://www.rcsb.org/docs/programmatic-access/file-download-services. PDBBind[34]: http://www.pdbbind.org.cn/. LBA[35]: https://zenodo.org/records/4914718. MSP[35]: https://zenodo.org/records/4962515. QM9[47]: https://springernature.figshare.com/ndownloader/files/3195389. The processed datasets are collected and available at https://doi.org/10.6084/m9.figshare.c.8197622. Source data are provided with this paper.

## Code availability

Our code is available at https://github.com/jiaor17/EPT with https://doi.org/10.5281/zenodo.17901948[68].

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

## Acknowledgements

This work was jointly supported by the following projects: the Fundamental and Interdisciplinary Disciplines Breakthrough Plan of the Ministry of Education of China (No. JYB2025XDXM101), the National Natural Science Foundation of China (No.62376276), the Beijing Nova Program (No. 20230484278), the Fundamental Research Funds for the Central Universities, and the Research Funds of Renmin University of China.

## Author contributions

R.J., X.K., and Z.Y. collected and preprocessed the datasets. R.J. and X.K. designed the backbone model and conducted the pretraining and downstream experiments. Z.Y., L.Z., F.R., and W.T. conducted the virtual screening experiment. R.J., X.K., L.Z., Z.Y., and W.H. wrote the manuscript. The study was supervised by W.H. and Y.L.

## Competing interests

The authors declare no competing interests.
