## [Transparent Peer Review file · Nature Communications]

An Equivariant Pretrained Transformer for Unified 3D Molecular Representation Learning

Corresponding Author: Professor Wenbing Huang

Version 0:

Reviewer comments:

Reviewer #1

(Remarks to the Author)

The Equivariant Pretrained Transformer (EPT) is a cross-domain foundation model for 3D molecules, leveraging E(3)-equivariant transformers and block-level denoising. Pretrained on 5.89M entries, it outperforms previous SOTA in ligand binding affinity, protein, and molecular property prediction. EPT also shows results in virtual screening, identifying anti-COVID-19 drugs and novel compounds with higher binding affinities.

The noteworthy results:

1. The manuscript includes a comprehensive list of baselines for comparison and evaluates their model's performance across three major downstream tasks.
2. A detailed ablation study is conducted, clearly highlighting the importance of each component in the proposed method.
3. The methods section is well-formulated and easy to follow, providing sufficient detail for reproducibility.
4. Beyond benchmarking baselines and proposing their method, the authors also perform virtual screening (3CL-protease) to address a real-world problem, demonstrating practical applicability.

Major issues that need addressing to improve clarity and depth:

1. Pre-training Mechanism: What is the purpose of pre-training? Is there explicit evidence to explain how pre-training enhances performance on downstream tasks? What is the relationship between pre-training and downstream tasks? Why were translation and rotation denoising chosen as the pre-training objectives?
2. Baseline Implementation: In Figure 2b, were all comparable methods re-trained by the authors? The manuscript does not mention this. How were the baseline methods implemented?
3. EPT-Scratch Performance: In Figure 2b, the EPT-Scratch variant already outperforms all other methods. Can the authors explain why? Since some baselines (e.g., Uni-Mol) also use transformer architectures, were the comparisons conducted fairly?
4. Atom3D Dataset Details: In line 151, the authors mention using the Atom3D dataset for the MSP task. Can more details about the dataset be provided? What are the labels, and how much data does it contain? Given the difficulty of obtaining crystal structures for single protein mutations, how does the model perform when predicting on less accurate or predicted structures?
5. Figure 3b Clarification: In Figure 3b, what does "Base" refer to? Does it indicate no pre-training?
6. Ranking Task and Virtual Screening:
 - 6.1 In Section 2.4, why were only 10 additional ligands sampled as negatives? In virtual screening scenarios, it is common to rank inhibitors among millions of molecules.
 - 6.2 The manuscript does not explain how negative samples were sampled until Figure 5, which mentions docking. What docking program was used?
 - 6.3 How were the Glide program and ESM+Uni-Mol implemented? Details of their re-implementation are lacking, which could impact performance if not used properly.
 - 6.4 In line 248, how were docking and MD simulation performed? How was the delta G of ~ -30 kcal/mol calculated? The process is unclear. While the supplementary material includes MD details, this should be referenced in the main manuscript, along with the total simulation time (e.g., how many nanoseconds were run for each simulation?). More details on baseline implementation, especially for Glide, are needed.
 - 6.5 For greater impact (Advice), the top candidates from the virtual screening should be validated through wet lab experiments to confirm their inhibitory potency.

The manuscript has several issues related to typos and clarity that need addressing:

1. In Figure 2b, the colors are difficult to distinguish, making comparisons challenging. Using a consistent color scheme for each method would improve readability.

The authors need to address the above points to strengthen the paper's contributions and ensure its impact on the field. By providing more details on the pre-training mechanism, baselines implementation, dataset characteristics and experimental procedures.

(Remarks on code availability)

I reviewed the source code on private GitHub and found the guide to be promising. It's structured in a way that allows me to execute the code step by step.

The source code includes the training component only, the author should also provide an inference module to enable others to replicate the results reported in the paper. This would significantly enhance the reproducibility and utility of the work.

Reviewer #2

(Remarks to the Author)

This manuscript presents Equivariant Pretrained Transformer (EPT), a novel all-atom foundation model designed for unified 3D molecular representation learning across multiple domains, including small molecules, proteins, and complexes. The authors leverage an E(3)-equivariant transformer architecture, which is capable of modelling atom-level information while incorporating block-level features. The model is pretrained using a block-level denoising task and evaluated on various downstream tasks, showcasing remarkable performance. Additionally, the model is applied to identify small-molecule drug candidates targeting 3CL protease, a critical target in SARS-CoV-2 replication.

Overall, the manuscript is well-structured and presents a significant advancement in the field of 3D molecular representation learning. EPT addresses a key gap in the literature by providing a unified framework for handling diverse molecular systems. The E(3)-equivariant transformer backbone is innovative and aligns well with the physics principle of molecular interactions. Notably, the block-level denoising task contributes to the model's ability to learn robust and generalizable representations.

However, there are several aspects of the study that could benefit from further clarification and discussion:

1. In Eq. (1), the edges are defined within a range of δ_{\max} . How does this definition reconcile with the fully-connected graph described in line 301? Is there a potential conflict between the two representations?

2. While the model demonstrates impressive performance on many tasks, its performance on the $\langle R^2 \rangle$ task in the QM9 dataset appears to be less satisfactory. Could the authors provide an explanation for this result?

3. In the virtual screening experiments, the fine-tuned model effectively identified known anti-COVID-19 drugs as top candidates. Could the authors further elaborate on the fine-tuning process?

4. Following question 3, it is worth noting that Xocova, one of the marketed anti-COVID-19 drugs, was ranked relatively low in the list. Could the authors provide a possible explanation for this outlier result?

Minor editing suggestions:

The title number in line 345 is redundant.

(Remarks on code availability)

Reviewer #3

(Remarks to the Author)

Summary

The paper introduces Equivariant Pretrained Transformer (EPT), a novel all-atom foundation model designed to leverage 3D molecular data across multiple scientific domains, including proteins, small molecules, and their complexes. Unlike previous approaches limited to single domains, EPT utilizes an E(3)-equivariant transformer architecture to capture both atom-level and block-level structural features. A key innovation is the use of a block-level denoising pretraining task, enabling better generalization.

EPT is trained on a large-scale dataset of 5.89 million 3D molecular structures, spanning diverse biological and chemical entities. Experimental results demonstrate EPT's strong performance on various downstream tasks, with state-of-the-art results in ligand binding affinity prediction and competitive outcomes in protein and molecular property predictions.

EPT also shows promise in drug discovery, notably ranking 7 out of 8 known anti-COVID-19 drugs among the top 200 candidates from 1,978 FDA-approved drugs targeting SARS-CoV-2's 3CL protease. Moreover, Molecular Dynamics simulations validate 7 newly identified compounds with superior binding affinities, highlighting EPT's potential for impactful therapeutic development.

Strengths

* The empirical performance of EPT is robust. The manuscript selects several key downstream tasks highly relevant to drug discovery (LBA, MSP, MPP, affinity prediction, ranking) and demonstrates that EPT consistently outperforms existing strong baseline methods. This indicates that EPT holds significant promise for enhancing efficiency in drug design workflows.

* The manuscript is well-structured and easy to follow, clearly presenting motivation and technical details. Figures are well-designed, clear, and visually appealing.

* The concept of unifying diverse 3D molecular representations within a single framework is valuable. This approach enhances data utilization efficiency and contributes a foundation model beneficial to the community, potentially reducing the necessity for extensive model selection.

Weaknesses

* Several critical discussions are missing, affecting the manuscript's rigor:

* EPT underperforms Frad on MPP tasks, and the paper claims that EPT-10 (enlarging the backbone from 6 to 10 layers) achieves state-of-the-art results. However, this comparison might be unfair since the network sizes for other models are fixed or unknown. What if the network sizes of other models are also increased? It would be beneficial to include an explanation for EPT-10 explicitly in the caption of Table 1.

* The claim that EPT is the first general model capable of handling atomic systems of various types (lines 59-60) may be overstated. AlphaFold3, for instance, also manages multiple types of 3D molecules. It is advisable to moderate this claim or provide a detailed comparison with relevant related work.

* The manuscript contains minor typographical errors:

* Line 13 is missing a space between "property prediction" and "[56]".

* Panel (d) appears to be missing in Figure 1.

* Several questions remain open for clarification:

* Could the authors discuss the key differences between various correlation metrics (RMSE, Pearson, Spearman) used in LBA evaluation? Clarifying the specific strengths and focus areas of these metrics would better contextualize the model's performance.

* Can the current block definitions be universally applied across general molecular systems? Currently, blocks are defined specifically for proteins and molecules—can this be extended to other molecular systems such as nucleic acids or crystals?

* Technically, this paper does not propose an entirely novel method. The E(3) Transformer architecture and pre-training strategy are pre-existing techniques. The primary novel contribution is the block-level representation for proteins and molecules, yet alternative block definitions are also possible (e.g., defining blocks as amino acids for proteins and heavy atoms for molecules is not a fundamentally new approach).

(Remarks on code availability)

Version 1:

Reviewer comments:

Reviewer #1

(Remarks to the Author)

I would like to thank the authors for their thorough responses to my concerns and questions, as they have fully addressed all the issues I raised. In the revision manuscript, the author have included extensive additional experimental details. Moreover, they validated the affinity of the top-ranked candidate through web-lab testing, which is particularly impressive to see in an AI-focused paper. Given these improvements, I believe the manuscript is now suitable for acceptance.

(Remarks on code availability)

Reviewer #3

(Remarks to the Author)

Q1: Regarding Supplementary Fig. 5, I am quite confused about why adding layers to Frad results in a significant performance drop on MPP. Please provide a reasonable discussion to explain this phenomenon. Typically, simply increasing model parameters without tuning the default training recipe leads to poor performance. To ensure a fair comparison, a reasonable hyperparameter search should be conducted for Frad. Otherwise, the current results are not convincing.

Q2: Regarding AF3, I agree that the objectives of EPT and AF3 diverge. However, my intention in raising this concern was to encourage a more comprehensive literature review on the unification of tasks for atomic systems—not just in representation learning, but also in broader understanding and generation tasks (e.g., [1,2,3]). Please include a discussion based on a more thorough literature review to better contextualize your contribution (not just AF3).

[1] Xia, Yingce, et al. "Nature Language Model: Deciphering the Language of Nature for Scientific Discovery." arXiv preprint arXiv:2502.07527 (2025).

[2] Zhang, Gongbo, et al. "Unigenx: Unified generation of sequence and structure with autoregressive diffusion." arXiv preprint arXiv:2503.06687 (2025).

[3] Lu, Shuqi, et al. "Uni-3dar: Unified 3D generation and understanding via autoregression on compressed spatial tokens." arXiv preprint arXiv:2503.16278 (2025).

Q3 & Q4: Thanks for your revision.

Q5: While I understand the distinction between Spearman and Pearson, a more detailed discussion on the difference between RMSE and Pearson should be provided. Both metrics evaluate the error between predicted and ground-truth binding affinities. So why use the two metrics for this task? Are there specific benefits, or is this choice simply based on precedent in existing work?

Q6: Thank you for including the experiment demonstrating extensibility to nucleic acids. However, since the block definition is based on domain knowledge, this should be noted as a limitation in the paper. EPT is not a "free lunch" when it comes to generalizing to new atomic systems. This manual effort should be acknowledged, as some systems may not have easily definable blocks—this constitutes a limitation of the method.

Q7: I agree that this paper makes meaningful technical contributions. However, in the response to Q7, it was stated: "We highlight three synergistic innovations..." but four innovations were listed. This appears to be a typo and should be corrected.

Overall Comments: Thank you to the authors for their hard work in addressing many of my concerns. Some issues have been satisfactorily resolved. However, several important concerns remain unaddressed, as outlined above.

(Remarks on code availability)

Reviewer #4

(Remarks to the Author)

The manuscript entitled "An Equivariant Pretrained Transformer for Unified 3D Molecular Representation Learning" written by Jiao et al. reported the development of Equivariant Pretrained Transformer (EPT), an atom-based pre-trained model capable of predicting ligand binding affinity with a performance reportedly better than previous state-of-the-art methods. To demonstrate the potential of EPT, they screened FDA-approved drugs against SARS-CoV-2 3CL protease, where they identified both known 3CL protease inhibitors and other novel candidates as their top-ranking hits. For validation, in the latest revision of this manuscript, they acquired top candidate Ac-Leu-Leu-Nle-CHO and experimentally evaluated it in a biochemical assay against SARS-CoV-2 3CL protease.

Regarding their wet-lab experiments, the identity and purity of the Ac-Leu-Leu-Nle-CHO compound that they purchased from a commercial vendor (MedChemExpress) was convincingly demonstrated by the mass spec and NMR spectra. The biochemical assay was then conducted by a separate CRO (Taoshu BioScience) using recombinant 3CL protease and a fluorogenic substrate, which is standard and typical for evaluating compounds as 3CL protease inhibitor. But I would recommend the authors to further strengthen their manuscript by addressing the following points as in a minor revision:

1. A known 3CL protease inhibitor (for example nirmatrelvir) should be included in the same assay as a positive inhibitor control to validate the assay setup and also benchmark the relative potency of Ac-Leu-Leu-Nle-CHO, as IC₅₀ values may vary with the assay conditions. The CRO should already have data in this regard as they establish and optimize the assay. Alternatively, the authors can cite IC₅₀ values of known 3CL inhibitors from paper that also used this CRO for the same biochemical assay.

2. 3CL protease inhibitor Nirmatrelvir exhibited an IC₅₀ value of 10 nM in a similar biochemical assay against SARS-CoV-2 3CL protease. But it was only ranked 173th by the EPT. In contrast, IC₅₀ value of their top-ranking candidate Ac-Leu-Leu-Nle-CHO is 5.47 μM. So Nirmatrelvir is >500-fold more potent than Ac-Leu-Leu-Nle-CHO. The authors should provide discussion to explain this mismatch to help readers understand what the true capability of this model is, as well as its potential limitations.

A minor typo is in the abstract that IC₅₀ is 5.47 M.

(Remarks on code availability)

Version 2:

Reviewer comments:

Reviewer #3

(Remarks to the Author)

I would recommend accept after very minor edits. The technical clarifications and new analyses directly address my earlier concerns. If the authors can incorporate the (optional) numeric ablation deltas for Frad's LR sweep in Supp. §1.7, that would further strengthen transparency (currently only the best results are recorded), but I would not hold the paper for this. Also in the Discussion, consider fixing a small grammatical glitch (“it block-aware, E(3)-equivariance nature...”) to “its block-aware, E(3)-equivariant nature...”.

(Remarks on code availability)

Reviewer #4

(Remarks to the Author)

I appreciate the efforts that the authors put together to further improve their manuscript. They have fully addressed all the minor points that I raised and have strengthened their manuscript accordingly. Therefore, I am more than glad to recommend publication of this manuscript. Congratulations!

(Remarks on code availability)

Point-by-Point Response to Reviewer Comments

We thank all the reviewers for their insightful feedback and constructive comments during this round of review. Here we provide detailed responses to address the proposed concerns and revise our paper accordingly. The reviewers' comments are written in blue, our responses are in black, and the changes to the manuscripts are in red.

Referee #1

Major issues that need addressing to improve clarity and depth:

Q1: Pretraining Mechanism: What is the purpose of pretraining? Is there explicit evidence to explain how pretraining enhances performance on downstream tasks? What is the relationship between pretraining and downstream tasks? Why were translation and rotation denoising chosen as the pretraining objectives?

We thank the reviewer for this insightful question regarding the motivation and impact of our pretraining strategy. We address your concerns point by point below.

A1.1: The purpose of pretraining.

As already presented in Lines 15-28, pretraining has become a widely successful technique in modern AI systems (e.g., ChatGPT), enabling learning from large-scale unlabeled data. In our framework, pretraining primarily aims to equip the model with a generalizable understanding of 3D molecular geometries across diverse domains, including small molecules, proteins, and their complexes. Given the scarcity of labeled data in molecular science, such as protein-ligand affinity scores, pretraining helps the model learn rich structural priors from abundant unlabeled 3D data.

A1.2: Empirical evidence on the efficacy of pretraining.

In Figure 3b/c, we demonstrate that all EPT variants pretrained on different domains (small molecules or proteins) consistently outperform the Base model without pretraining, on both the Atom3D and QM9 datasets. Notably, multi-domain pretraining yields stronger performance than single-domain pretraining models (i.e. Molecule-Only and Protein & Complex Only) across all downstream tasks, demonstrating effective cross-domain knowledge transfer.

We have revised the caption of Figure 3 by adding: "Base" refers to the EPT model without pretraining. This clarification emphasizes that the experiments in Figure 3 demonstrate the efficacy of pretraining.

A1.3: The relationship between pretraining and downstream tasks.

Pretraining establishes a general-purpose 3D representation that captures atom- and block-level interactions under $E(3)$ symmetry. Downstream tasks such as property prediction or binding affinity estimation rely on accurate modeling of such interactions, which are implicitly learned during pretraining. Our use of equivariant architectures ensures that the learned features are physically consistent and transferable.

A1.4: The use of translation and rotation denoising.

We design our pretraining task to match the physical nature of molecular systems. Specifically, we apply random translation and rotation perturbations to blocks, and train the model to recover them. This choice is motivated by several considerations:

- (1) Geometric Awareness: The training objective requires the model to learn $E(3)$ -equivariant denoising terms for reconstructing original conformations. This approach explicitly enables the model to not only understand the geometry of input systems, but also learn to stabilize noisy systems through rotation and translation.
- (2) Hierarchical Nature: The pretraining objectives are applied to entire blocks, rather than individual atoms, allowing the model to better capture hierarchical and contextual relationships.
- (3) Empirical Superiority: As shown in Figure 3c, the combined block-level translation and rotation denoising strategy outperforms both atom-level and translation-only denoising strategies, indicating that our chosen objective is optimal across molecular scales.

We have revised Lines 18-21 and 71-76 to reflect the above points.

Q2: Baseline Implementation: In Figure 2b, were all comparable methods re-trained by the authors? The manuscript does not mention this. How were the baseline methods implemented?

We sincerely thank the reviewer for raising this important point regarding the baseline implementations, which is crucial for ensuring the validity of our comparisons.

In Figure 2b, we did not re-train the baseline models but utilized the performance metrics reported in the original publications of ProtNet [1] and ProFSA [2], following the standard practice for the well-established LBA benchmark. This approach ensures consistency and avoids potential discrepancies from reimplementing/re-training, which can be highly resource-intensive and non-trivial due to dependencies and version-specific implementation details, potentially introducing unintended variations. The LBA benchmark, along with its standard data splits (ID30/ID60), is widely recognized and used in the field. Following the precedent set by prior works, including [1] and [2] themselves, which also report results

relative to established baselines using published numbers, our setting allows for direct and fair comparison.

We acknowledge the reviewer's concern for thoroughness and want to assure them that this methodology is chosen specifically to uphold the integrity and comparability of our results. Should the reviewer deem it essential, we would be willing to discuss the feasibility of re-running specific baselines as an additional analysis, though we believe the current approach provides the most consistent and reliable comparison foundation.

We have mentioned this in Lines 145-147 of the main paper.

[1] Wang, Limei, et al. "Learning Hierarchical Protein Representations via Complete 3D Graph Networks." The Eleventh International Conference on Learning Representations.

[2] Gao, Bowen, et al. "Self-supervised Pocket Pretraining via Protein Fragment-Surroundings Alignment." The Twelfth International Conference on Learning Representations.

Q3: EPT-Scratch Performance: In Figure 2b, the EPT-Scratch variant already outperforms all other methods. Can the authors explain why? Since some baselines (e.g., Uni-Mol) also use transformer architectures, were the comparisons conducted fairly?

We thank the reviewer for this insightful observation. The superior performance of EPT-Scratch compared to prior methods like Uni-Mol is largely attributed to architectural differences: Uni-Mol adopts a dual-tower model to encode the ligand and pocket separately, whereas EPT uses a unified model that jointly encodes the entire protein-ligand complex. This unified model enables more effective learning of cross-molecular interactions, even without pretraining. These results actually align with our motivation of developing a unified model, which better captures the geometric nature of molecular complexes and enhances downstream performance.

We have provided more explanations in Lines 155-160.

Q4: Atom3D Dataset Details: In line 151, the authors mention using the Atom3D dataset for the MSP task. Can more details about the dataset be provided? What are the labels, and how much data does it contain? Given the difficulty of obtaining crystal structures for single protein mutations, how does the model perform when predicting on less accurate or predicted structures?

We thank the reviewer for this question and sorry for the insufficient details. The MSP dataset in Atom3D is based on SKEMPI 2.0 [1], which contains experimentally measured binding affinities (Kd) of protein-protein complexes before and after single-point mutations. Each mutation is labeled as 1 if it improves binding (mutant Kd < wild-type Kd), and 0 otherwise. The dataset includes 893 positive and 3255 negative examples, and is split by 30% sequence identity for evaluation.

To generate 3D mutant structures, the authors of Atom3D applied in silico mutagenesis using *PyRosetta*, where local sidechain repacking was performed within 10 Å of the mutated residue. This modeling protocol introduces predicted structures based on experimentally resolved wild-type PDB entries, rather than experimentally determined mutant structures. As such, while the protocol is standardized and widely used, it may introduce some structural inaccuracies. Nevertheless, our model demonstrates robustness in this setting.

We added the details of MSP data construction in Lines 168-172.

[1] Jankauskaitė, Justina, et al. "SKEMPI 2.0: an updated benchmark of changes in protein–protein binding energy, kinetics and thermodynamics upon mutation." *Bioinformatics* 35.3 (2019): 462-469.

Q5: Figure 3b Clarification: In Figure 3b, what does "Base" refer to? Does it indicate no pretraining?

We thank the reviewer for pointing this out. In Figure 3b, "Base" refers to the EPT model without any pretraining, i.e., trained from scratch using only the downstream task data.

We refined the caption of Figure 3b with additional details.

Q6: Ranking Task and Virtual Screening:

Q6.1: In Section 2.4, why were only 10 additional ligands sampled as negatives? In virtual screening scenarios, it is common to rank inhibitors among millions of molecules.

We appreciate this thoughtful question and apologize for any confusion. In Section 2.4, we used just 10 additional ligands as negatives to efficiently train EPT to discriminate true ligands from decoys. Importantly, once trained, EPT can in principle rank inhibitors among millions of molecules.

In the evaluation process, we limited the negative ligands to 10 per target across all 500 targets. While we recognize that virtual screening typically involves ranking binders from much larger compound libraries, scaling to millions of molecules across 500 targets would be computationally prohibitive. Crucially, our further analysis here shows that the evaluation metric remains stable whether using 10 or more decoys per target (Supplementary 1.10), demonstrating that this reduced setting still enables robust performance comparisons across targets.

Supplementary Fig. 7: $avg_rank(n)$ and $\Delta avg_rank(n) = avg_rank(n) - avg_rank(n+1)$ over the number of negative samples n on the id30 split. The Oracle method indicates positive samples are always ranked as top-1.

To complement our large-scale benchmark, we conducted a real-world virtual screening experiment targeting SARS-CoV-2 3CL protease. In this evaluation, we ranked true binders among thousands of FDA-approved drugs, and our model successfully ranked 3 out of 8 known positive drugs within the top 10 candidates. Notably, two additional top-10 candidates - Saquinavir and Conivaptan HCl - have been independently reported as 3CL protease inhibitors in prior studies [1,2]. As detailed in our response to Q6.5, we further validated these findings by synthesizing and testing Ac-Leu-Leu-Nle-CHO (ranked 7th), which showed the most favorable binding energy (MM/PBSA ΔG) among the top candidates. Experimental validation confirmed its inhibitory activity with an IC_{50} of 5.47 μM . These results collectively demonstrate EPT's practical utility in real-world drug screening applications involving large compound libraries.

We have revised Lines 249-254 to address the reviewers' concern.

[1] Chiou, Wei-Chung, et al. "Repurposing existing drugs: identification of SARS-CoV-2 3C-like protease inhibitors." *Journal of enzyme inhibition and medicinal chemistry* 36.1 (2021): 147-153.

[2] Drayman, Nir, et al. "Drug repurposing screen identifies masitinib as a 3CLpro inhibitor that blocks replication of SARS-CoV-2 in vitro." *BioRxiv* (2020): 2020-08.

Q6.2: The manuscript does not explain how negative samples were sampled until Figure 5, which mentions docking. What docking program was used?

Sorry for the confusion. We use the Glide program for docking negative samples. To address your concerns, we have explained this point in Line 237.

Q6.3: How were the Glide program and ESM+Uni-Mol implemented? Details of their re-implementation are lacking, which could impact performance if not used properly.

Thank you for this valuable suggestion. We provide more details as follows:

(1) Glide:

We used Schrödinger Suite 2021-2 for docking. Protein structures were preprocessed with PrepWizard by adding hydrogen atoms and optimizing with the OPLS3 force field at pH 7.4. Ligands were prepared using LigPrep, preserving the original chirality, while Epik was used to predict pKa values and generate protonation states at pH 7.0. Ligand conformations were optimized with the S-OPLS force field, and a single conformation was retained for docking. The receptor grid was generated with an innerbox size of $10 \times 10 \times 10 \text{ \AA}$ and an outerbox defined as $(x_{\max} - x_{\min} + 20) \times (y_{\max} - y_{\min} + 20) \times (z_{\max} - z_{\min} + 20)$, with the force field set to OPLS3 and other parameters left at default values. Docking was then carried out using Glide SP (standard precision).

(2) ESM+Uni-Mol:

We appreciate the reviewer's concern regarding the implementation details of the ESM+Uni-Mol baseline. In our experiments, we used the docked-PDBBind dataset to ensure consistency across all methods. For the protein representation, we adopted the pretrained model `esm2_t33_650M_UR50D` [1], generating an embedding of shape $(R, 1280)$, where R denotes the number of amino acid residues in the protein. We then computed the protein-level embedding by averaging the residue-wise embeddings. For the ligand representation, we employed the `UniMolRepr` function from the `unimol_tools` [2,3] package to obtain a 512-dimensional ligand embedding.

To combine the two modalities, we utilized a standard dual-tower architecture. Specifically, for each protein-ligand pair, we concatenated the protein and ligand embeddings as the joint input. This concatenated vector was passed through a Feed Forward Network (FFN) with 4 hidden layers and 1 output layer, where the hidden size is 512 and the activation function is SiLU. The model was trained with a learning rate of $5e-4$, and all remaining setups, including the training objective and negative rates, were kept identical to those used in our EPT model.

The above details have been explained in Supplementary Information Section 1.3.1. We hope these clarifications adequately address the reviewer's concern.

[1] Lin, Z., Akin, H., Rao, R., Hie, B., Zhu, Z., Lu, W., ... & Rives, A. (2022). Language models of protein sequences at the scale of evolution enable accurate structure prediction. *BioRxiv*, 2022, 500902.

[2] Zhou, G., Gao, Z., Ding, Q., Zheng, H., Xu, H., Wei, Z., ... & Ke, G. Uni-Mol: A Universal 3D Molecular Representation Learning Framework. In *The Eleventh International Conference on Learning Representations*.

[3] Gao, Z., Ji, X., Zhao, G., Wang, H., Zheng, H., Ke, G., & Zhang, L. (2023). Uni-qsar: an auto-ml tool for molecular property prediction. *arXiv preprint arXiv:2304.12239*.

Q6.4: In line 248, how were docking and MD simulation performed? How was the delta G of ~-30 kcal/mol calculated? The process is unclear. While the supplementary material includes MD details, this should be referenced in the main manuscript, along with the total simulation time (e.g., how many nanoseconds were run for each simulation?). More details on baseline implementation, especially for Glide, are needed.

We thank the reviewer for pointing out the need for additional details regarding the docking and MD simulation procedures.

(1) Docking and MD simulation:

Molecular docking was performed using Glide. The detailed protocol is provided in our response to Q6.3 and has also been incorporated into Section 1.3.1 of the Supplementary Information.

As for MD simulation, we have carefully reviewed Section 1.4.2 in the Supplementary Information and added important implementation details as: **This was followed by a restrained constant number of particles, volume, and temperature (NVT) ensemble equilibration for 100 ps, and a constant number of particles, pressure, and temperature (NPT) ensemble equilibration for an additional 100 ps.**

In addition, we included a summary of the MD simulation procedure, including the total simulation time, in Lines 269-276 as: **Detailed protocols for molecular docking, MD simulations, and ΔG calculation are provided in the Supplementary Information Section 1.4.2 and 1.4.3. The SARS-CoV-2 3CLpro-ligand complexes were simulated using GROMACS 2024.2 [1]. Energy minimization was performed using the steepest descent algorithm, followed by 100 ps of restrained equilibration under the constant number of particles, volume, and temperature (NVT) ensemble and 100 ps under the constant number of particles, pressure, and temperature (NPT) ensemble. Subsequently, 100 ns of unrestrained production MD simulations were carried out at 310 K and 1 bar.**

(2) Binding free energy calculation:

Binding free energy (ΔG , e.g., ~-30 kcal/mol) was estimated using the MM/PBSA approach based on explicit-solvent MD trajectories, considering the complex, receptor, and ligand components [1-2]. Further details regarding the calculation of ΔG are provided in Section 1.4.3 of the Supplementary Information.

[1] Valdés-Tresanco, Mario S., et al. "gmx_MMPBSA: a new tool to perform end-state free energy calculations with GROMACS." *Journal of chemical theory and computation* 17.10 (2021): 6281-6291.

[2] King, Edward, et al. "Recent developments in free energy calculations for drug discovery." *Frontiers in molecular biosciences* 8 (2021): 712085.

Q6.5: For greater impact (Advice), the top candidates from the virtual screening should be validated through wet lab experiments to confirm their inhibitory potency.

We thank the reviewer for the valuable suggestion regarding the experimental validation. In the development pipeline of SARS-CoV-2 3CL protease inhibitors, we first performed EPT-based virtual screening, and identified 12 candidate compounds. Subsequent MD simulations combined with MM/PBSA analysis revealed two hits: Ac-Leu-Leu-Nle-CHO ($\Delta G = -29.34$ kcal/mol) and Saquinavir ($\Delta G = -31.08$ kcal/mol), both exhibiting significantly more favorable binding free energies than the positive control Leritrelvir ($\Delta G = -6.16$ kcal/mol).

Notably, Saquinavir was reported to inhibit 3CL protease with an IC₅₀ of approximately 9.92 μM [1], providing additional support for its potential. Accordingly, we further synthesized and experimentally evaluated Ac-Leu-Leu-Nle-CHO, which demonstrated an IC₅₀ of approximately 5.47 μM . These wet lab experiments confirm the inhibitory potency of the top candidates selected by our EPT. The experimental details are as follows.

(1) Compound information:

The compound Ac-Leu-Leu-Nle-CHO (also known as MG-101; CAS No. 110044-82-1) was synthesized and purchased from MedChemExpress (MCE, Cat# HY-18964). The compound was supplied with a reported purity of $\geq 98\%$ and was used without further purification. Its identity and purity were confirmed by MCE based on Mass Spectrometry (MS) and Nuclear Magnetic Resonance (NMR) analysis (see Supplementary Fig. 3), as documented in the Certificate of Analysis (CoA). The CoA and additional quality control documents are available upon request.

Supplementary Fig. 3: Analytical data for Ac-Leu-Leu-Nle-CHO. **a**, ESI-MS spectrum showing the $[M+H]^+$ peak at m/z 384.2, consistent with the expected molecular weight (383.28). **b**, 1H NMR spectrum (DMSO- d_6) confirming the chemical structure and purity of the compound. **Source:** Certificate of Analysis provided by MedChemExpress.

(2) SARS-CoV-2 3CL protease inhibition assay:

The inhibitory activity of Ac-Leu-Leu-Nle-CHO against SARS-CoV-2 3CL protease was evaluated by TaoShu Bioscience (Shanghai, China). The compound was prepared as a 10 mM stock in DMSO and serially diluted using a 10-point, 3-fold dilution series starting from 5 mM. Diluted compound (80 nL per well, transferred by Echo) was dispensed into 384-well plates (Corning 4514), with each well adjusted to a final volume of 8 μ L (top dose 50 μ M, final DMSO 1%).

Recombinant SARS-CoV-2 3CLpro was diluted in assay buffer (50 mM HEPES pH 7.5, 2 mM DTT, 0.01% Triton X-100, 0.01% BSA) to 20 nM, and 4 μ L enzyme was pre-incubated with compound at room temperature for 30 minutes. The reaction was initiated by addition of 4 μ L of a fluorogenic peptide substrate (HiLyte Fluor 488-ESATLQSGLRKAK(QXL520)-NH₂, final 2 μ M).

After 60 minutes at room temperature, fluorescence (Ex/Em 480/540 nm) was measured using an EnVision plate reader (PerkinElmer). Percent inhibition was calculated relative to positive and negative controls. IC₅₀ values were determined by four-parameter logistic fitting (4PL, XLfit Model 205, Microsoft Excel), and dose-response curves were plotted in Python (matplotlib) using the fitted parameters. All experiments were performed in duplicate (n = 2).

Supplementary Fig. 4: Dose-response curve of Ac-Leu-Leu-Nle-CHO in the SARS-CoV-2 3CL protease inhibition assay. The absolute IC₅₀ value was determined to be 5.47 μ M using a 4PL model. The curve was plotted in Python (matplotlib) with fitted parameters provided in the experimental report.

Overall, the content related to the Ac-Leu-Leu-Nle-CHO assay has been added to Lines 286–289 of our paper and to Section 1.5 of the Supplementary Information.

[1] Chiou, Wei-Chung, et al. "Repurposing existing drugs: identification of SARS-CoV-2 3C-like protease inhibitors." *Journal of enzyme inhibition and medicinal chemistry* 36.1 (2021): 147-153.

Q7. In Figure 2b, the colors are difficult to distinguish, making comparisons challenging. Using a consistent color scheme for each method would improve readability.

We thank the reviewer for the suggestion. We revised **Figure 2b** to use a consistent color scheme for each method, thereby improving readability. Inspired by this comment, we also refined the **molecular diagrams in Figures 1d and 4b** to further enhance the quality of presentation.

Q8. The source code includes the training component only, the author should also provide an inference module to enable others to replicate the results reported in the paper. This would significantly enhance the reproducibility and utility of the work.

We thank the reviewer for this suggestion. The evaluation codes are provided in the “/evaluation” folder of our codebase. We also list the commands required to run the evaluation, which are integrated into the training pipelines for each downstream task.

Referee #2

There are several aspects of the study that could benefit from further clarification and discussion:

Q1: In Eq. (1), the edges are defined within a range of δ_{\max} . How does this definition reconcile with the fully-connected graph described in line 301? Is there a potential conflict between the two representations?

We thank the reviewer for the insightful observation. The term "fully-connected graph" refers to the fact that all atom pairs are considered in the self-attention mechanism, ensuring that long-range dependencies can be modeled across the entire molecular structure.

In contrast, the edge types defined in Eq. (1) with distance thresholds are specifically used to introduce localized edge-aware attention, allowing the model to pay extra attention to local and near-local interactions. These edge encodings are incorporated as part of the edge attention term (denoted R_{ij}) in the attention computation (see Eq. (10–11)), alongside the self-attention and distance-based attention components.

Therefore, there is no contradiction: our architecture supports global communication via a fully-connected attention graph, while explicit edge features are used to encode and emphasize local interactions to further improve expressiveness.

We have provided more explanations of the setting in Lines 357-360.

Q2: While the model demonstrates impressive performance on many tasks, its performance on the $\langle R^2 \rangle$ task in the QM9 dataset appears to be less satisfactory. Could the authors provide an explanation for this result?

We appreciate the reviewer's attention to the $\langle R^2 \rangle$ results on the QM9 dataset. We acknowledge that the performance of EPT on this particular target is less competitive compared to other properties.

This observation aligns with prior studies [1,2], which have highlighted that $\langle R^2 \rangle$ —the electronic spatial extent—can suffer from negative transfer in pretrained models. In addition, we note that some previous works adopt a specific projection head tailored for $\langle R^2 \rangle$, based on the formulation $R^2 = \sum_{i=1}^N e_i \|x_i - x_m\|_2^2$, where e_i is a predicted scalar and x_m is the molecular center-of-mass (CoM). Inspired by this finding, we also apply an additional projection head to EPT, achieving a test MAE of 0.037, which is significantly better than the result obtained with a standard regression head (0.122).

[1] Jiao, Rui, et al. "Energy-motivated equivariant pretraining for 3d molecular graphs." Proceedings of the AAAI Conference on Artificial Intelligence. Vol. 37. No. 7. 2023.

[2] Feng, Shikun, et al. "Fractional denoising for 3d molecular pretraining." International Conference on Machine Learning. PMLR, 2023.

Q3: In the virtual screening experiments, the fine-tuned model effectively identified known anti-COVID-19 drugs as top candidates. Could the authors further elaborate on the fine-tuning process?

We thank the reviewer for requiring more details regarding the fine-tuning process used in the virtual screening experiments. The pretrained EPT model was fine-tuned on the Docked-PDBBind dataset (described in *Methods – Datasets*), where each protein receptor is associated with 1 positive ligand and 10 negative decoys. We formulated two tasks on this dataset—affinity prediction and ranking (see Section 2.4). For virtual screening on the 3CL protease, we employed the model weights fine-tuned under the ranking task.

During training, each epoch iterates over all protein receptors in the dataset. For each receptor, a protein-ligand pair is sampled based on a hyperparameter `negative_rate` $\in [0, 1]$. In particular, a negative ligand is uniformly selected with probability `negative_rate`, while a positive ligand is selected with probability `1-negative_rate`. In the affinity prediction task, `negative_rate` is fixed to 0 to train only on positive complexes. For the ranking task, the `negative_rate` is set as described in *Supplementary Table 8*.

Architecturally, we treated EPT as an encoder that generates a graph-level representation for each protein-ligand complex. Different output heads are tailored for different tasks. For the affinity prediction task, we employ the same head as in Eq. (24), which outputs a single scalar representing the predicted binding affinity. For the ranking task, we appended a 2-layer Feedforward Neural Network (FFN) as a task-specific output head, where the hidden size is the same as that of EPT and the activation function is ReLU. It outputs a probability vector $[y_p, y_n]$, where y_p, y_n indicate the probabilities of the ligand being positive or negative, respectively. If the ligand is a negative sample, the target label is $[0, 1]$, otherwise the label is defined as $[y, 1 - y]$, where $y = (pK/16.0)^{0.183}$. Here, pK is the ground-truth binding score, 16.0 is the maximum pK observed in the training set (used for normalization), and 0.183 is an empirical exponent reflecting the prior that a complex with pK = 9.0 corresponds to a 0.9 positive probability. The cross-entropy loss is leveraged as the training objective.

We performed finetuning under three data splits: id30, id60, and scaffold. The virtual screening against the 3CL protease specifically used the model finetuned under the id60 split.

We hope this clarifies the fine-tuning procedure used in our study.

Q4: Following question 3, it is worth noting that Xocova, one of the marketed anti-COVID-19 drugs, was ranked relatively low in the list. Could the authors provide a possible explanation for this outlier result?

We thank the reviewer for bringing this phenomenon to our attention. Xocova adopts a non-peptidic triazine scaffold, which is markedly different from the peptidic or peptidomimetic structures of mainstream 3CL protease inhibitors such as Leritrelvir, Rupintrivir, 3CLpro-1, GC376, Simnotrelvir, Nirmatrelvir, and Lufotrelvir (see *Supplementary Fig. 1*). This

observation indicates that, although the model exhibits strong discriminative power for mainstream structures, its performance still has room for improvement when it comes to identifying and evaluating compounds with novel scaffolds.

Supplementary Fig. 1: EPT rankings of marketed anti-COVID-19 drugs. Global rankings of eight marketed anti-COVID-19 drugs based on EPT scores, along with their 2D chemical structures.

We have added the explanation regarding the outlier Xocova to Lines 260-261 of our paper and to Section 1.4.1 of the Supplementary Information.

Minor editing suggestions:

Q5: The title number in line 345 is redundant.

We thank the reviewer for pointing this out. We have removed the redundant title number.

Referee #3

Several critical discussions are missing, affecting the manuscript's rigor:

Q1: EPT underperforms Frad on MPP tasks, and the paper claims that EPT-10 (enlarging the backbone from 6 to 10 layers) achieves state-of-the-art results. However, this comparison might be unfair since the network sizes for other models are fixed or unknown. What if the network sizes of other models are also increased? It would be beneficial to include an explanation for EPT-10 explicitly in the caption of Table 1.

We thank the reviewer for this helpful suggestion. To assess the impact of model size, we have conducted additional experiments by scaling both Frad and EPT from 6 to 12 layers. Results on HOMO, LUMO, and Gap Prediction show that Frad does not benefit consistently from larger network size, while EPT continues to improve up to 10–12 layers, indicating better scalability and stability.

To clarify the use of EPT-10 in Table 1, we have modified the caption of Table 1 as follows: We also include the performance for both Frad and EPT when calling the networks from 6 to 12 layers in the Supplementary Fig. 5. Frad does not benefit consistently from larger network size, while EPT continues to improve up to 10–12 layers, indicating better scalability and stability.

Supplementary Fig. 5: Comparison of performance of EPT and Frad under increasing layers. Mean absolute errors (MAE) on HOMO, LUMO, and Gap prediction are reported for both EPT and Frad with varying numbers of layers (6, 8, 10, and 12). EPT demonstrates consistent performance improvements as the model depth increases. In contrast, Frad does not consistently benefit from deeper architectures.

Q2: The claim that EPT is the first general model capable of handling atomic systems of various types (lines 59-60) may be overstated. AlphaFold3, for instance, also manages multiple types of 3D molecules. It is advisable to moderate this claim or provide a detailed comparison with relevant related work.

We thank the reviewer for pointing this out. We agree that the original wording may overstate the novelty of our approach. We provide a detailed comparison between AlphaFold3 and our method here.

While AlphaFold3 and EPT share the common motivation of modeling diverse molecular domains, they differ in scope and application. AlphaFold3 focuses on structure prediction, whereas EPT is designed as a general-purpose representation learning framework. Through self-supervised pretraining, EPT learns transferable representations that support a broader range of downstream tasks beyond structural modeling alone, including molecular property prediction and protein-ligand affinity scoring.

We have moderated the claim and revised the wording in Lines 61–65 and Lines 305-310.

The manuscript contains minor typographical errors:

Q3: Line 13 is missing a space between "property prediction" and "[56]".

Q4: Panel (d) appears to be missing in Figure 1.

We thank the reviewer for pointing these out, and have fixed the typos and missing panel numbers in the revised manuscript.

Several questions remain open for clarification:

Q5: Could the authors discuss the key differences between various correlation metrics (RMSE, Pearson, Spearman) used in LBA evaluation? Clarifying the specific strengths and focus areas of these metrics would better contextualize the model's performance.

We thank the reviewer for the suggestion. In the LBA task, we report RMSE, Pearson, and Spearman to provide a comprehensive evaluation from different perspectives. The specific strengths and focus areas of these metrics are as follows.

RMSE (Root Mean Square Error) measures the absolute error between predicted and ground-truth binding affinities. It is sensitive to large errors and directly reflects how close the model's predictions are with respect to the ground-truth pKds.

Pearson Correlation Coefficient captures the linear correlation between predicted and ground-truth values. It is scale-invariant and focuses on whether high-affinity complexes are consistently predicted with higher scores. In practice, a high Pearson correlation is valuable when the relative magnitude of predicted affinities is more important than the precise values.

Spearman Correlation Coefficient evaluates the rank-order alignment between predictions and labels, focusing only on the order rather than the values themselves.

We have extended the details of evaluation metrics in Lines 433-442.

Q6: Can the current block definitions be universally applied across general molecular systems? Currently, blocks are defined specifically for proteins and molecules—can this be extended to other molecular systems such as nucleic acids or crystals?

We thank the reviewer for this thoughtful question. In our current work, blocks are defined based on domain knowledge: amino acid residues for proteins and heavy-atom groups for small molecules, reflecting domain-specific structural and chemical units.

However, the underlying framework of EPT is flexible and extensible. Once domain-specific block definitions are provided, the same architecture can be applied to other molecular systems. To directly prove this, we extended the EPT block vocabulary to nucleic acids and evaluated its zero-shot generalization on Nucleic acid-ligand binding prediction. Specifically, we introduced four deoxyribonucleic acid and four ribonucleic acid building blocks into the vocabulary while keeping the model architecture unchanged. The model was finetuned on Protein-Protein, Protein-Nucleic acid, and Protein-Ligand complexes from the PDBBind dataset and then evaluated on novel data type: Nucleic acid–Ligand complexes. As shown in Table 1 below, the pretrained model achieves significantly better performance compared to training from scratch, demonstrating that EPT generalizes effectively to nucleic acid systems with simple vocabulary extensions. **These results are added as Section 1.6 in the revised Supplementary Information.**

Table 2: The mean and standard deviations of 3 runs on the PDBBind NL complexes.

Model	Pearson \uparrow	Spearman \uparrow
EPT-pretrain	0.402 \pm 0.037	0.315 \pm 0.058
EPT-scratch	0.359 \pm 0.057	0.263 \pm 0.061

Q7: Technically, this paper does not propose an entirely novel method. The E(3) Transformer architecture and pretraining strategy are pre-existing techniques. The primary novel contribution is the block-level representation for proteins and molecules, yet alternative block definitions are also possible (e.g., defining blocks as amino acids for proteins and heavy atoms for molecules is not a fundamentally new approach).

We sincerely thank the reviewer for this insightful observation, which helps clarify the scope of our contributions. We agree that the foundational concepts of E(3)-equivariance and transformer architectures are established previously, and our work builds thoughtfully upon these shoulders. The core novelty of EPT lies not in inventing wholly new technical components, but in their purposeful integration into the first unified framework that achieves high-performance, all-atom representation learning across diverse molecular domains (small molecules, proteins, complexes) at scale. We highlight three synergistic innovations enabling this:

- (1) Unified Block-Level Representation: We abstract both proteins and molecules into a consistent format, enabling a single transformer model to operate across different molecular modalities including small molecules, proteins, and complexes. While block definitions like residues or heavy-atom groups are domain-informed, our contribution is to integrate them seamlessly into a unified representation framework.

- (2) **Model Architecture Enhancements:** The EPT backbone is not a direct reuse of existing E(3)-equivariant transformers (e.g. TorchMD-Net and Equiformer). We introduce several key modifications to make it suitable for all-atom, multi-domain representation learning, including block-aware embedding (Eqs. (5-8)), hybrid scalar-vector updations (Eqs. (3-4)), and distance-aware attention bias (Eqs. (10-11)). Moreover, unlike conventional E(3)-equivariant transformers that rely on GNN-based edge updates, EPT employs a pure attention mechanism that maintains compatibility with standard transformer optimizations (e.g., xFormers for memory efficiency) while preserving equivariance. This design allows computationally tractable integration of geometric biases. Implementation details are provided through PyTorch-style pseudocode in Appendix 1.6.1. As demonstrated in Appendix 1.6.2, EPT achieves superior memory efficiency compared to TorchMD-Net and Equiformer across varying system sizes (up to 2048 nodes). This scalability enables modeling of large molecular systems without sacrificing representational capacity.
- (3) **Block-Level Pretraining Objective:** We design a block-level translation and rotation denoising task that leverages rigid-body perturbations to capture local geometry and context. This task is specifically aligned with the physics of molecular systems, and we show in Figure 3c that it outperforms the atom-level denoising counterparts.
- (4) **All-Atom Multi-Domain Pretraining:** EPT is pretrained on large-scale datasets across three domains (molecules, proteins, complexes) and shows strong performance on diverse downstream tasks. This supports our claim that EPT is a general-purpose 3D molecular representation model.

We hope the above explanations sufficiently address the reviewer's concern.

Point-by-Point Response to Reviewer Comments

We thank all the reviewers for their insightful feedback and constructive comments during this round of review. Here we provide detailed responses to address the proposed concerns and revise our paper accordingly. The reviewers' comments are written in blue, our responses are in black, and the changes to the manuscripts are in red.

Referee #1

I would like to thank the authors for their thorough responses to my concerns and questions, as they have fully addressed all the issues I raised. In the revision manuscript, the author have included extensive additional experimental details. Moreover, they validated the affinity of the top-ranked candidate through web-lab testing, which is particularly impressive to see in an AI-focused paper. Given these improvements, I believe the manuscript is now suitable for acceptance.

We sincerely thank the reviewer for the thoughtful and encouraging comments. We appreciate the recognition of the expanded experimental details and the wet-lab validations. We are grateful for the time and constructive feedback that helped strengthen the manuscript.

Referee #3

Q1: Regarding Supplementary Fig. 5, I am quite confused about why adding layers to Frad results in a significant performance drop on MPP. Please provide a reasonable discussion to explain this phenomenon. Typically, simply increasing model parameters without tuning the default training recipe leads to poor performance. To ensure a fair comparison, a reasonable hyperparameter search should be conducted for Frad. Otherwise, the current results are not convincing.

We thank the reviewer for raising this concern. To address it, we additionally performed a targeted hyperparameter search for Frad by sweeping learning rates across $\{1e-4, 5e-5, 1e-5\}$ and reporting the best result. We included more clarification on this setting in Supplementary 1.7. This alleviated, but did not eliminate the performance drop observed on MPP as depth increases. The results are updated in Supplementary Fig. 5 and also provided as below. We believe this degradation is tied to the TorchMD-Net backbone used by Frad. For TorchMD-Net, vector features are initialized to zero and gradually updated via gated interactions with scalar filters and edge directions. Such zero initialization could promote the collapse of vector features across nodes as depth increases, owing to the over-smoothing issue in message passing neural networks [1]. In contrast, our backbone explicitly initializes non-zero equivariant vectors at the embedding stage via neighbor-weighted positional differences (Eq. 8), so anisotropic cues exist from the first layer. We also fuse scalar and vector features with a GVP-style FFN and a layer-normalized

gate (Eqs. 13–15), which stabilizes vector norms and curbs accumulation or bleed-through of low-contrast signals across many layers. Consistent with this mechanism, EPT scales more robustly with the number of layers.

[1] Rusch, T. Konstantin, Michael M. Bronstein, and Siddhartha Mishra. "A Survey on Oversmoothing in Graph Neural Networks." SAM Research Report 2023 (2023).

Q2: Regarding AF3, I agree that the objectives of EPT and AF3 diverge. However, my intention in raising this concern was to encourage a more comprehensive literature review on the unification of tasks for atomic systems—not just in representation learning, but also in broader understanding and generation tasks (e.g., [1,2,3]). Please include a discussion based on a more thorough literature review to better contextualize your contribution (not just AF3).

[1] Xia, Yingce, et al. "Nature Language Model: Deciphering the Language of Nature for Scientific Discovery." arXiv preprint arXiv:2502.07527 (2025).

[2] Zhang, Gongbo, et al. "Unigenx: Unified generation of sequence and structure with autoregressive diffusion." arXiv preprint arXiv:2503.06687 (2025).

[3] Lu, Shuqi, et al. "Uni-3dar: Unified 3D generation and understanding via autoregression on compressed spatial tokens." arXiv preprint arXiv:2503.16278 (2025).

We appreciate the reviewer's suggestion to broaden the contextualization of unified modeling for atomic systems beyond AF3. In the revision, we added a dedicated discussion in the Supplementary 1.11 to survey three complementary directions, including structure-aware generalist encoders, unified 3D generative frameworks, and sequence-level foundation models. We also rewrote Line 308-316 in the discussion section with a concise summary positioning EPT within this broader landscape.

Q5: While I understand the distinction between Spearman and Pearson, a more detailed discussion on the difference between RMSE and Pearson should be provided. Both metrics evaluate the error between predicted and ground-truth binding affinities. So why use the two metrics for this task? Are there specific benefits, or is this choice simply based on precedent in existing work?

We thank the reviewer for this valuable suggestion. The two metrics capture complementary aspects of performance. **RMSE** measures absolute calibration error in affinity and thus directly reflects how accurate the predicted magnitudes are. In contrast, **Pearson** (Correlation Coefficient) quantifies linear association and is invariant to affine transformations of the predictions. Therefore, it is insensitive to systematic offsets or scale mismatches. Reporting these two metrics allows us to measure both linear ranking (Pearson) and scale fidelity (RMSE), which are both relevant for binding-affinity applications. Plus, our choice also follows established practice in existing benchmarks and prior work [1], so we keep both metrics for comparability. We have highlighted the reference in Line 432.

[1] Townshend, Raphael John Lamarre, et al. "ATOM3D: Tasks on Molecules in Three Dimensions." Thirty-fifth Conference on Neural Information Processing Systems Datasets and Benchmarks Track.

Q6: Thank you for including the experiment demonstrating extensibility to nucleic acids. However, since the block definition is based on domain knowledge, this should be noted as a limitation in the paper. EPT is not a “free lunch” when it comes to generalizing to new atomic systems. This manual effort should be acknowledged, as some systems may not have easily definable blocks—this constitutes a limitation of the method.

We appreciate the reviewer’s constructive point and agree that EPT is currently not a “free lunch” when extending to new atomic systems. Our block-level denoising assumes an externally pre-defined decomposition of atoms into blocks, such as residues for proteins or phosphate–sugar–base units for nucleic acids, which depends on domain knowledge. We have made this limitation explicit in Line 319-322 and also outlined future directions such as data-driven block discovery that could reduce manual effort in domains where block definitions are ambiguous.

Q7: I agree that this paper makes meaningful technical contributions. However, in the response to Q7, it was stated: “We highlight three synergistic innovations...” but four innovations were listed. This appears to be a typo and should be corrected.

We thank the reviewer for catching the inconsistency in our previous response. This was a typo that we meant to state “four synergistic innovations”, and the list indeed contained four items. We apologize for the confusion.

Referee #4

1. A known 3CL protease inhibitor (for example nirmatrelvir) should be included in the same assay as a positive inhibitor control to validate the assay setup and also benchmark the relative potency of Ac-Leu-Leu-Nle-CHO, as IC₅₀ values may vary with the assay conditions. The CRO should already have data in this regard as they establish and optimize the assay. Alternatively, the authors can cite IC₅₀ values of known 3CL inhibitors from paper that also used this CRO for the same biochemical assay.

We thank the reviewer for this important suggestion, and apologize for missing the report of the positive control in our previous response. The CRO included GC376 (ranked #82 by EPT) as a positive inhibitor control under the same assay conditions. Using a 4-parameter logistic (4PL) fit to the dose–response curves, the absolute IC₅₀ values were 9.54 nM for GC376, aligning well with previous reported value of 30 nM [1] and thus showcasing the reliability of the experimental settings. These control data validate the assay setup and provide a potency benchmark for our candidate. We have added the curves to the Supplementary Fig. 4, which is also shown as below.

[1] Ma, Chunlong, et al. "Boceprevir, GC-376, and calpain inhibitors II, XII inhibit SARS-CoV-2 viral replication by targeting the viral main protease." *Cell research* 30.8 (2020): 678-692.

2. 3CL protease inhibitor Nirmatrelvir exhibited an IC₅₀ value of 10 nM in a similar biochemical assay against SARS-CoV-2 3CL protease. But it was only ranked 173th by the EPT. In contrast, IC₅₀ value of their top-ranking candidate Ac-Leu-Leu-Nle-CHO is 5.47 μM. So Nirmatrelvir is >500-fold more potent than Ac-Leu-Leu-Nle-CHO. The authors should provide discussion to explain this mismatch to help readers understand what the true capability of this model is, as well as its potential limitations.

We appreciate the reviewer's comment. Our screening pipeline relies on docked complexes. Target pockets and candidate ligands are first docked with Glide, and EPT then re-scores these complexes. Empirically, EPT ranks most truly active compounds near the top. Nevertheless, the docking software is not always accurate, while high-fidelity affinity prediction requires precise structures. Consequently, the ranking cannot guarantee that the most potent ligand will always appear at the very top. We further evaluated undervalued molecules in Supplementary Fig. 1, and found that Nirmatrelvir (EPT rank #173) and

Lufotrelvir (EPT rank #176) have Glide ranks #218 and #203, respectively. EPT partially corrects Glide's ordering but cannot fully recover from suboptimal inputs. We present this as a pipeline-based limitation, and note that improvements in the upstream docking software are expected to enhance the final performance. We thank the reviewer for this valuable suggestion once again, and have included the above discussion in Supplementary 1.4.1.

A minor typo is in the abstract that IC50 is 5.47 M.

Thanks for pointing this out! We have corrected this typo.z g